# DUALCONTRAST: UNSUPERVISED DISENTANGLING OF CONTENT AND TRANSFORMATIONS WITH IMPLICIT PARAMETERIZATION

## ABSTRACT

Unsupervised disentanglement of content and transformation is significantly important for analyzing shape focused scientific image datasets, given their efficacy in solving downstream image-based shape-analyses tasks. The existing relevant works address the problem by explicitly parameterizing the transformation latent codes in a generative model, significantly reducing their expressiveness. Moreover, they are not applicable in cases where transformations can not be readily parametrized. An alternative to such explicit approaches is contrastive methods with data augmentation, which implicitly disentangles transformations and content. However, the existing contrastive strategies are insufficient to this end. Therefore, we developed a novel contrastive method with generative modeling, DualContrast, specifically for unsupervised disentanglement of content and transformations in shape focused image datasets. DualContrast creates positive and negative pairs for content and transformation from data and latent spaces. Our extensive experiments showcase the efficacy of DualContrast over existing self-supervised and explicit parameterization approaches. With DualContrast, we disentangled protein composition and conformations in cellular 3D protein images, which was unattainable with existing disentanglement approaches.

## 1 INTRODUCTION

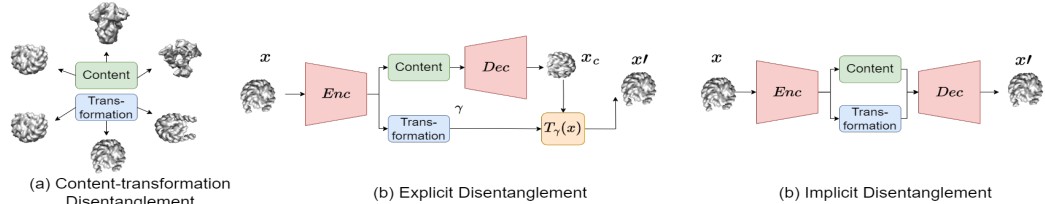

(a) Content-transformation Disentanglement    (b) Explicit Disentanglement    (b) Implicit Disentanglement

Figure 1: (a) The concept of content-transformation disentanglement, whereas changing the content changes the protein identity, and changing transformation changes the state of the protein. Explicit methods (b) use the transformation space to infer a fixed parameter set, whereas **our** implicit method (c) do not restrict the transformations to a fixed set of parameters. Figure uses toy protein images for visualization, they are not used for experiments in this form.

In many real-world image datasets, particularly in scientific imaging domains, the object shapes being visualized may undergo multiple transformations (Skafte & Hauberg, 2019; Bepler et al., 2019). These shape focused image datasets can thus be regarded as samples generated from two independent factors, one representing the semantic attribute termed content and the other representing the transformations. Content is regarded as the *form* of an object being visualized in the image (Skafte & Hauberg, 2019) that is unaltered after applying any nuisance transformation. On the other hand, transformation dictates the specific *state* or *realization* of that form of the object. Taking proteins inside the cells as an example, changing the transformation factor changes the conformation of the protein where the protein identity (*i.e.*, protein composition) does not change (Fig. 1). On the other hand, changing the content factor is analogous to changing the proteins from one identity to another

through compositional change. Such a phenomenon holds for many shape focused image datasets. Disentangling content and transformation factors of images can significantly facilitate several downstream shape analysis tasks, including shape clustering (Levy et al., 2022b), alignment (Uddin et al., 2022; Bepler et al., 2019), zero-shot transfer (Zhou et al., 2020), bias removal (Ngweta et al., 2023; Lee et al., 2021), cross-domain retrieval (Kahana & Hoshen, 2022; Piran et al., 2024), etc. (Liu et al., 2022).

A number of works have been done to directly or indirectly solve the content-transformation disentangling problem (Jha et al., 2018; Kahana & Hoshen, 2022; Cosmo et al., 2020; Skafte & Hauberg, 2019; Uddin et al., 2022; Bepler et al., 2019), whereas many of them use annotated factors for training. However, such annotation is hard to obtain, and thus, unsupervised disentanglement methods are desired. Only a few works (Skafte & Hauberg, 2019; Bepler et al., 2019; Uddin et al., 2022) have addressed the problem in a completely unsupervised setting. Among them, (Skafte & Hauberg, 2019) performed explicit parameterization of transformation codes as diffeomorphic transformations and achieved noteworthy disentanglement of content and transformation in several image datasets. (Bepler et al., 2019) and (Uddin et al., 2022) focus on representing known transformation types as transformation codes. (Bepler et al., 2019) performed disentanglement of two-dimensional translation and rotation, whereas (Uddin et al., 2022) demonstrated disentanglement of a broader set of transformations that can be explicitly parameterized.

Though these unsupervised explicit parameterization methods have demonstrated several successes, they have several major limitations in general: (1) These method uses the transformation codes to generate only a few parameterized transformations. In reality, many transformations do not have a well-defined parameterized form, e.g., protein conformation changes, viewpoint change (LineMod), etc. These explicit methods can not disentangle such transformation by design. (2) They use Spatial Transformer Networks (STN) (Jaderberg et al., 2015) for inferring the transformation codes, which requires a continuous parameterization of the transformation to be disentangled. The continuity of many transformations (*e.g.*, SO(3) rotation) in neural networks is often a concern (Zhou et al., 2019). Moreover, (3) these methods infer spurious contents during disentanglement in real scenarios where transformations not predefined are present in the dataset (Fig. 3). Due to these limitations, an unsupervised content-transformation disentangling method without explicit transformation representation is much desired.

Consequently, in this work, we focused on the more generalized and unique setting of unsupervised content-transformation disentanglement without any explicit parameterization of the transformation code. Previously, a theoretical study by Von Kügelgen et al. (2021) showed that standard Contrastive Learning (CL) methods, e.g., SimCLR (Chen et al., 2020), can disentangle content from style, where style can be regarded as the transformations used for data augmentation. However, these popular CL methods (Chen et al., 2020) have rarely achieved any disentanglement in practice (Ngweta et al., 2023), and their ability in disentangling transformations other than those used for augmentation is not well-explored. Moreover, we found that SimCLR with geometric data augmentation can not disentangle any transformation well in our shape focused image datasets of interest (Table 1).

To this end, we develop a novel generative contrastive learning method for implicit content-transformation disentanglement, termed "DualContrast" (Fig. 2). Unlike existing contrastive strategies, it creates "positive" and "negative" contrastive pairs w.r.t. both content and transformation latent codes. We found that several shape transformations in image datasets lie closely in the normally distributed latent space that is designed to capture in-plane rotation information in a contrastive manner. Consequently, we create negative pairs w.r.t. transformation and positive pairs w.r.t. content by simply rotating the samples. Without prior knowledge, creating positive pairs w.r.t. the transformation codes directly from the input samples is not possible. To this end, we generate samples from an identical distribution of the transformation representations in the embedding space and treat them as positive pairs for the transformation codes. Given our observation, this encourages the disentanglement of several shape transformations other than rotation in our dataset. In addition, we randomly permute the samples in the batch to create a negative pair for a batch of samples w.r.t. content. Upon creating both "positive" and "negative" pairs w.r.t. both codes, we penalize similarity between negative pairs and distance between positive pairs for each code. Finally, we incorporate these contrastive losses in a Variational Autoencoder (VAE) based architecture with VAE reconstruction loss and train the model in an end-to-end manner.

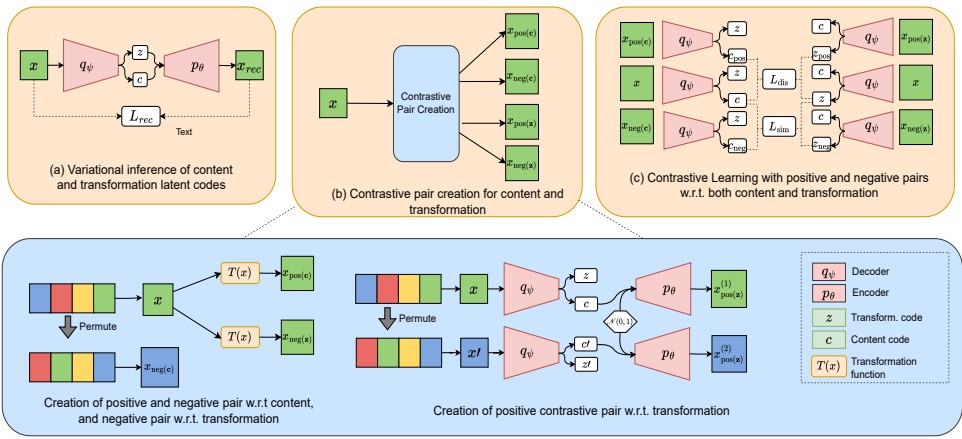

Figure 2: Our proposed contrastive learning-based unsupervised content-transformation disentanglement pipeline. (a) The variational inference of content and transformation codes with $L_{VAE}$. (b) Contrastive pair creation strategy for content and transformation codes. The process is delineated in the bottom panel. Additional visualization available in Appendix Fig. 9. (c) Contrastive losses. In DualContrast, the contrastive pair creations and reconstruction happen simultaneously, and the encoder and decoder network are optimized with both contrastive and reconstruction losses in each iteration.

With such a strategy, by just performing simple rotation to create contrastive pairs, DualContrast disentangles a much wider range of transformations caussing subtle changes in the pixel space, e.g., viewpoint in Linemod, several writing styles in MNIST, etc. With DualContrast, we, for the first time, solved the scientific problem of disentangling protein identities from protein conformations (defined as subtle changes) in simulated 3D protein images (Fig. 5 and Fig. 6) inside the cell, known as cellular cryo-ET subtomograms. DualContrast could successfully disentangle the protein identities as content and protein conformations as transformation in simulated subtomograms, which is not achievable with previous methods. Overall, our qualitative and quantitative experimental results show the efficacy of DualContrast in isolating content from transformation compared to existing relevant methods.

A summary of our contributions is as follows:

- We develop a novel contrastive generative modeling strategy that creates positive-negative pairs for both content and transformation latent codes. We introduced a novel way of designing contrastive pairs for transformation from both data and latent space.

- We show that DualContrast is effective in disentangling a wide range of transformations that causes subtle changes in pixel space, from the content in various shape focused image datasets.

- Leveraging DualContrast, we, for the first time, disentangled protein identities (*i.e.* composition) from protein conformations in simulated 3D cellular cryo-ET subtomograms, as a proof of principle. DualContrast is the first method that can identify distinct conformations of proteins from protein mixture subtomogram datsets.

## 2   RELATED WORKS

**Unsupervised Content-Transformation Disentanglement:** There are a few (Skafte & Hauberg, 2019; Uddin et al., 2022; Bepler et al., 2019; Von Kügelgen et al., 2021) approaches that deal with the unsupervised disentanglement of content and transformation. However, these methods perform explicit parameterization of the transformation codes as affine or some specific parameterized transformation. In contrast, our method does not impose any explicit parameterization on the transformation latent codes and does not face the issues of the explicit methods (Section 1).

**Contrastive Learning based Disentanglement:** Contrastive learning is the primary building block of our method. Contrastive Learning with data augmentation with existing approaches, *e.g.* , InfoNCE loss-based SimCLR (Chen et al., 2020), etc., has been previously used by (Von Kügelgen et al., 2021; Kahana & Hoshen, 2022) to isolate content from style. Though in several scenarios,

style in shape focused images can be referred to as transformations, these works do not specifically focus on shape focused real-world images. Moreover, our experiments demonstrate that SimCLR with rotation augmentation in a standard encoder-decoder framework leads to very poor disentanglement of content and transformation (Table 1). Also, (Von Kügelgen et al., 2021) only creates positive pairs w.r.t content and negative pair w.r.t. style with data augmentation in SimCLR. We have observed this scheme does not work well in our scenario. Unlike these works, our approach uses a novel strategy to create "negative" and "positive" pairs for both content and transformation of latent codes without any InfoNCE loss.

**Unsupervised Content-style disentanglement:** Apart from (Von Kügelgen et al., 2021), several methods (Ngweta et al., 2023; Ren et al., 2021b;a; Liu et al., 2021; Kwon & Ye, 2021; Wang et al., 2023) exist that perform unsupervised content-style disentanglement, focusing on natural images. Unlike these methods, our work primarily focuses on disentangling content and transformation in shape focused image datasets. Moreover, we do not depend on any ImageNet pretrained models as our images of interest differ greatly from the natural images of ImageNet. The very recent work by Ngweta et al. (2023) assumes access to the style factors to disentangle that style from content in feature outputs from pretrained models. Unlike this work, we do not assume access to the transformations beforehand for disentanglement.

**Protein Composition-Conformation Disentanglement:** One of the major contributions of our work is disentangling protein identity characterized by large compositional variability as the content codes and protein states characterized by conformational variability or subtle compositional variability as transformation codes from 3D cellular cryo-electron tomography (cryo-ET) subimages or subtomograms. Previously, Harmony (Uddin et al., 2022) was used to disentangle large compositional variability in cellular subtomograms. HEMNMA-3D (Harastani et al., 2021) and TomoFlow (Harastani et al., 2022) were used to disentangle subtle compositional or conformational variability in subtomograms using known templates. SpatialVAE (Bepler et al., 2019) disentangled subtle conformational variability from 2D cryo-EM images. Ours is the first work to deal with both large and subtle compositional variability in two different latent spaces for 3D cryo-ET subtomograms.

Further discussions on the related works can be found in Appendix. Section A.1.

## 3 METHOD

### 3.1 DISENTANGLING CONTENT AND TRANSFORMATION

Disentangling content and transformation refers to learning one mapping $h : \mathcal{X} \to \mathcal{C} \times \mathcal{Z}$, where $\mathcal{X}$ is an input space, $\mathcal{C}$ and $\mathcal{Z}$ are the intermediary content and transformation representation spaces respectively. Here, $h$ can be decomposed as $h_c : \mathcal{X} \to \mathcal{C}$ and $h_z : \mathcal{X} \to \mathcal{Z}$ representing the content and transformation mapping, respectively. In an unsupervised setting, distinguishing between content and transformation can be tricky. In this paper, we distinguish between content and transformation with respect to a family of transformations $\mathcal{T} : \mathcal{X} \to \mathcal{X}$ such that the following two conditions hold:

- **Condition 1:** $\forall\, T \in \mathcal{T}$ and $\forall\, \mathbf{x} \in \mathcal{X},\, h_c(T(\mathbf{x})) = h_c(\mathbf{x})$.
- **Condition 2:** $\exists\, T \in \mathcal{T}$ and $\forall\, \mathbf{x}^{(1)}, \mathbf{x}^{(2)} \in \mathcal{X},\, h_z(T(\mathbf{x}^{(1)})) = h_z(\mathbf{x}^{(2)})$.

**Condition 1** defines content space which is invariant w.r.t. $\mathcal{T}$. On the other hand, **condition 2** states that there is a $T \in \mathcal{T}$ that can make the transformation of any two samples equal when applied to any one of them. Thus it defines transformation space which is informative of $\mathcal{T}$.

For content-transformation (**c-z**) disentangling, changes in $\mathcal{C}$ must be unaffected by changes in $\mathcal{Z}$ and vice versa. In our problem setting, no labels associated with content space $C$ or transformation space $Z$ are known. The exact family of transformation $\mathcal{T}$ is also not known beforehand. Thus, we aim to simultaneously learn the family of transformations $\mathcal{T}$ and the content factors unaltered by $\mathcal{T}$.

### 3.2 METHOD OVERVIEW AND NOTATION

Our method is built upon a variational auto-encoder (VAE) based architecture (Fig. 2), where we perform inference of the content and transformation factors in the data. Consider, a sample space

$\mathcal{X}$, two latent distribution spaces $\Phi_Z$ and $\Phi_C$, two latent spaces $\mathcal{Z}$ and $\mathcal{C}$. We apply an encoder network $q_\psi : \mathcal{X} \to \mathcal{C} \times \mathcal{Z}$ on a sample space $bx \in \mathcal{X}$ to encode content distribution $\phi_c \in \Phi_c$ and transformation distribution $\phi_z \in \Phi_z$, where $(\phi_c, \phi_z) = q_\psi(\mathbf{x})$. Latent parameters $\mathbf{c} \in \mathcal{C}$, $\mathbf{z} \in \mathcal{Z}$ are drawn from distributions $\phi_c$ and $\phi_z$ respectively. These parameters are then decoded using a decoder network $p_\theta : \mathcal{C} \times \mathcal{Z} \to \mathcal{X}$ to produce $x_{\text{recon}} = p_\theta(\mathbf{c}, \mathbf{z})$.

## 3.3 VARIATIONAL INFERENCE OF CONTENT AND TRANSFORMATION CODES

Similar to a variational auto-encoder (VAE) setting, the encoder and decoder network parameters are simultaneously optimized to maximize the variational lower bound to the likelihood $p(\mathbf{x})$ called the evidence lower bound (ELBO). ELBO maximizes the log-likelihood of the data and minimizes the KL divergence between latent distributions and a prior distribution. The objective function for this can be written as:

$$
\begin{aligned}
p(\mathbf{x}) \geq & \mathbb{E}_{q_\psi(\mathbf{c}, \mathbf{z}|\mathbf{x})}[\log \frac{p_\theta(\mathbf{x}, \mathbf{c}, \mathbf{z})}{q_\psi(\mathbf{c}, \mathbf{z}|\mathbf{x})}] \\
= & \mathbb{E}_{q_\psi(\mathbf{c}, \mathbf{z}|\mathbf{x})} \log p_\theta(\mathbf{x}|\mathbf{c}, \mathbf{z}) - KL(q_\psi(\mathbf{c}|\mathbf{x})||p(\mathbf{c})) - KL(q_\psi(\mathbf{z}|\mathbf{x})||p(\mathbf{z})).
\end{aligned}
\tag{1}
$$

The log-likelihood term is maximized using a reconstruction loss $L_{\text{recon}}$. The prior for $\mathbf{c}$ and $\mathbf{z}$, $p(\mathbf{c})$ and $p(\mathbf{z})$ respectively are assumed to be standard normal distribution. In short, we write the KL terms as $L_{\text{KL}(c)}$ and $L_{\text{KL}(z)}$. To regularize the effect of prior on the final objective separately for two latent codes, we scale the two losses with two hyperparameters $\gamma_c$ and $\gamma_z$. Our loss function for variational inference becomes:

$$
L_{\text{vae}} = L_{\text{rec}} + \gamma_c L_{\text{KL}(\mathbf{c})} + \gamma_z L_{\text{KL}(\mathbf{z})}.
\tag{2}
$$

We observe that by deactivating the KL term associated with any one of the codes, the likelihood tends to be optimized through the other latent code. For instance, if $\gamma_c$ is set as 0 and $\gamma_z$ has a large value, then the $L_{\text{rec}}$ is optimized through $\mathbf{c}$ only. As a consequence, $\mathbf{c}$ captures all the information in $\mathbf{x}$, whereas $\mathbf{z}$ becomes totally independent of the data. In such a scenario, $\mathbf{z}$ will be a random code with no relation to the transformation of the data, which is not desirable.

## 3.4 CREATING CONTRASTIVE PAIR WITH RESPECT TO CONTENT

We adopt a contrastive strategy for $\mathbf{c}$-$\mathbf{z}$ disentanglement. We first develop a strategy to impose a structure to the content latent code with contrastive learning. To this end, we create negative and positive pairs for each data sample w.r.t. the content code. Hence, we randomly pick two samples $\mathbf{x}$ and $\mathbf{x}_{\text{neg}(\mathbf{c})}$, where $\mathbf{x}, \mathbf{x}_{\text{neg}(\mathbf{c})} \in \mathcal{X}$ and regard them as negative pairs of each other w.r.t the content code. Such strategy is commonly used in traditional self-supervised learning approaches (Chen et al., 2020; He et al., 2020). Considering the high heterogeneity present in the dataset, evidenced by a large number of classes, this approach accounts for the likelihood that any two randomly selected samples will have different content. To generate positive pairs w.r.t. the content code, we pick any sample $\mathbf{x}$ and transform it with a transformation $T \in \mathcal{T}$ to generate $\mathbf{x}_{\text{pos}(\mathbf{c})}$. We use the encoder $q_\psi$ to generate content codes $\mathbf{c}$, $\mathbf{c}_{\text{pos}}$, and $\mathbf{c}_{\text{neg}}$ respectively from $\mathbf{x}$, $\mathbf{x}_{\text{pos}(\mathbf{c})}$, and $\mathbf{x}_{\text{neg}(\mathbf{c})}$. We use a contrastive loss to penalize the distance between $\mathbf{c}$ and $\mathbf{c}_{\text{pos}}$ and the similarity between $\mathbf{c}$ and $\mathbf{c}_{\text{neg}}$. Our contrastive loss for content latent code can be written as: $L_{\text{con}(\mathbf{c})} = L_{\text{dist}}(\mathbf{c}, \mathbf{c}_{\text{pos}}) + L_{\text{sim}}(\mathbf{c}, \mathbf{c}_{\text{neg}})$. To implement $L_{\text{dist}}$ and $L_{\text{sim}}$, we use mean absolute cosine distance and mean absolute cosine similarity, respectively.

## 3.5 CREATING CONTRASTIVE PAIR WITH RESPECT TO TRANSFORMATION

Imposing contrastive loss on the content code satisfies condition 1 of the $\mathbf{c}$-$\mathbf{z}$ disentanglement (Section 3.1). However, the $\mathbf{c}$-$\mathbf{z}$ disentanglement remains only partial since it does not satisfy condition 2. To solve this issue, we design another contrastive loss w.r.t. the transformation code. Designing a contrastive loss that explicitly enforces condition 2 is not possible, given the transformation code that may equalize the transformation of two samples can not be approximated. To this end, we design a contrastive loss that implicitly encourages condition 2. We validated the design experimentally in Section 4.

The negative pair of a sample $\mathbf{x} \in \mathcal{X}$ w.r.t. transformation is generated while creating a positive pair of it w.r.t content. This is because a transformation $T$ on $\mathbf{x}$ alters its transformation, but not its contents. So, we regard $\mathbf{x}_{\text{pos}(\mathbf{c})}$ as $\mathbf{x}_{\text{neg}(\mathbf{z})}$. However, creating positive pairs of samples w.r.t. transformation is very challenging as they can not be created in the input space, unlike others. To this end, we adopted a strategy of creating pairs of samples from the same distribution in the transformation embedding space. We randomly sample $\mathbf{z}^{(1)}$ and $\mathbf{z}^{(2)}$ from $\mathcal{N}(0,1)$ and use the decoder $p_\theta$ to generate samples $\mathbf{x}_{\text{pos}(\mathbf{z})}^{(1)}$ and $\mathbf{x}_{\text{pos}(\mathbf{z})}^{(2)}$ from them (Fig. 2 (c)). In the decoder, as content input, we use content codes $\mathbf{c}^{(1)}$ and $\mathbf{c}^{(2)}$ obtained by encoding two random samples $\mathbf{x}^{(1)}$ and $\mathbf{x}^{(2)}$. Consequently, $\mathbf{x}_{\text{pos}(\mathbf{z})}^{(1)} = p_\theta(\mathbf{c}^{(1)}, \mathbf{z}^{(1)})$ and $\mathbf{x}_{\text{pos}(\mathbf{z})}^{(2)} = p_\theta(\mathbf{c}^{(2)}, \mathbf{z}^{(2)})$ serve as the positive pairs w.r.t. transformation. We then use the encoder $q_\psi$ to generate $\mathbf{z}$, $\mathbf{z}_{\text{neg}}$, $\mathbf{z}_{\text{pos}}^{(1)}$, and $\mathbf{z}_{\text{pos}}^{(2)}$ from $\mathbf{x}$, $\mathbf{x}_{\text{neg}(\mathbf{z})}$, $\mathbf{x}_{\text{pos}(\mathbf{z})}^{(1)}$, and $\mathbf{x}_{\text{pos}(\mathbf{z})}^{(2)}$ respectively. We use a contrastive loss to penalize the distance between $\mathbf{z}_{\text{pos}}^{(1)}$ and $\mathbf{z}_{\text{pos}}^{(2)}$ and similarity between $\mathbf{z}$ and $\mathbf{z}_{\text{neg}}$. Our contrastive loss for transformation can be written as: $L_{\text{con(z)}} = L_{\text{dist}}(\mathbf{z}_{\text{pos}}^{(1)}, \mathbf{z}_{\text{pos}}^{(2)}) + L_{\text{sim}}(\mathbf{z}, \mathbf{z}_{\text{neg}})$. Similar to $L_{\text{con(c)}}$, we use mean absolute cosine distance and mean absolute cosine similarity to implement $L_{\text{dist}}$ and $L_{\text{sim}}$, respectively.

*Why rotation?:* We used rotation as a transformation to create contrastive pairs because only rotation provided generalization to other shape-based transformations in the dataset, compared to other transformations such as translation, scale, blur, or color-based transformations (Details in Appendix A.2.3).

**Objective Function:** In summary, we train the encoder and decoder networks by simultaneously minimizing the loss components. Our overall objective function to minimize is:

$$L = L_{\text{vae}} + L_{\text{con(c)}} + L_{\text{con(z)}} \tag{3}$$

We optimize $L_{\text{vae}}$ for both $\mathbf{x}$ and the transformed image $\mathbf{x}_{\text{pos}(\mathbf{c})}$ (same as $\mathbf{x}_{\text{neg}(\mathbf{z})}$), which yielded better experimental result.

# 4 EXPERIMENTS & RESULTS

Table 1: Quantitative Results of unsupervised Content-Transformation Disentangling Methods. Among these methods, SpatialVAE and Harmony put constraints on the $\mathbf{z}$ code dimension given their explicit parameterization, others do not. The std. dev. over model training by setting 3 different random seeds remains within $\pm 0.04$. $D_{\text{score}}$ is written as $D$. For $D(c|c)$ and $D(z|z)$, higher is better. For $D(c|z)$ and $D(z|c)$, lower is better. For $SAP(c)$ and $SAP(z)$, higher is better.

| | MNIST | | | LineMod | | | Protein Subtomogram | | | | | |
| Method | $D(c\|c)$ | $D(c\|z)$ | $SAP(c)$ | $D(c\|c)$ | $D(c\|z)$ | $SAP(c)$ | $D(c\|c)$ | $D(c\|z)$ | $SAP(c)$ | $D(z\|z)$ | $D(z\|c)$ | $SAP(z)$ |
|---|---|---|---|---|---|---|---|---|---|---|---|---|
| SpatialVAE | 0.81 | 0.28 | 0.53 | 0.95 | 0.32 | 0.63 | 0.69 | 0.93 | 0.24 | 0.71 | 0.57 | 0.14 |
| Harmony | 0.82 | 0.31 | 0.51 | 0.90 | 0.56 | 0.34 | 0.95 | **0.01** | **0.94** | 0.52 | 0.90 | **0.38** |
| SimCLR (Discriminative) | 0.58 | 0.60 | 0.02 | 0.62 | 0.40 | 0.22 | 0.39 | 0.69 | 0.30 | 0.51 | **0.52** | 0.01 |
| SimCLR (Generative) | 0.53 | 0.67 | 0.14 | 0.61 | 0.79 | 0.18 | 0.53 | 0.59 | 0.06 | 0.55 | 0.54 | 0.01 |
| VAE with 2 latent space | 0.63 | 0.63 | 0.0 | 0.87 | 0.87 | 0.0 | 0.79 | 0.82 | 0.03 | 0.85 | 0.85 | 0.0 |
| VITAE | 0.77 | 0.32 | 0.45 | 0.92 | 0.90 | 0.02 | - | - | - | - | - | - |
| DualContrast (w/o $L_{\text{con(c)}}$) | 0.87 | **0.21** | **0.66** | 0.86 | **0.31** | **0.55** | 0.93 | 0.81 | 0.13 | 0.78 | 0.81 | 0.03 |
| DualContrast (w/o $L_{\text{con(z)}}$) | 0.79 | 0.85 | 0.06 | 0.79 | 0.86 | 0.07 | 0.99 | 0.86 | 0.13 | 0.86 | 0.63 | 0.23 |
| DualContrast | **0.89** | 0.31 | 0.58 | **0.95** | 0.48 | 0.47 | **1.00** | 0.56 | 0.44 | **0.86** | 0.64 | 0.22 |

**Benchmark Datasets:** We report results on MNIST (LeCun, 1998), LineMod (Hinterstoisser et al., 2013), and a realistically simulated protein subtomogram dataset. Only LineMod features RGB images among these datasets, while the others consist of grayscale images. The subtomogram is a 3D volumetric dataset, whereas the others are 2D images. For a detailed discussion of the datasets, we refer to the Appendix.

**Baselines:** We used the unsupervised **c**-**z** disentanglement methods, SpatialVAE (Bepler et al., 2019), Harmony (Uddin et al., 2022), VITAE (Skafte & Hauberg, 2019) with explicit parameterization and InfoNCE loss based standard self-supervised learning method with rotation augmentation (Von Kügelgen et al., 2021) as baseline approaches. Several generic disentangled representation learning methods, such as beta-VAE, Factor-VAE, beta-TCVAE, DIP-VAE, etc., are available that do not aim to disentangle specific factors such as content or transformation in the data but rather

aim to disentangle all factors of variation. However, previous studies (Bepler et al., 2019; Skafte & Hauberg, 2019) have shown that such methods perform worse than methods specifically aiming to disentangle content and transformation. Therefore, we did not use these methods as baselines in our experiments.

**Implementation Details:** We implemented the encoder using a Convolutional Neural Network (CNN) and the decoder using a Fully Connected Network (FCN). Our experiments used the same latent dimension for content and transformation codes, except for Harmony (Uddin et al., 2022) and SpatialVAE (Bepler et al., 2019), where a specific dimension needs to be used for transformation codes. We train our models for 200 epochs with a learning rate of 0.0001 in NVIDIA RTX A500 GPUs and AMD Raedon GPUs. We optimize the model parameters with Adam optimizer.

The hyperparameters $\gamma_z$ and $\gamma_c$ controls how close the $\mathbf{z}$ and $\mathbf{c}$ distribution will be to $\mathcal{N}(0, I)$. We found setting a small value $\approx 0.01$ for both provides optimal results in our experiments. Further details of the overall implementation are provided in the Appendix A.3.

**Evalutation:** We evaluate the methods based on two criteria: (1) the informativeness of the predicted $\mathbf{c}$ and $\mathbf{z}$ codes w.r.t the ground truth $\mathbf{c}$ and $\mathbf{z}$ factors, (2) the separateness of of the predicted $\mathbf{c}$ and $\mathbf{z}$ codes. We performed these evaluations both quantitatively and qualitatively.

For quantitative evaluations, there exists several metrics, *e.g.* , MIG score, $D_{\text{score}}$, SAP score, etc. Locatello et al. (2019) demonstrated that these metrics are highly correlated. To this end, we only used $D_{\text{score}}$ and SAP score to measure the disentanglement. $D_{\text{score}}$ is simply a measurement of how well the ground truth (gt) factors can be predicted from the corresponding latent codes. In our scenario, there are four such quantities - (a) predictivity of content gt given $\mathbf{c}$ code $D_{\text{score}}(c|c)$ or $\mathbf{z}$ code $D_{\text{score}}(c|z)$ (b) predictivity of transformation gt given $\mathbf{c}$ code $D_{\text{score}}(z|c)$ or $\mathbf{z}$ code $D_{\text{score}}(z|z)$. For MNIST and LineMod, we do not have any transformation gt, so we only reported values for (a). Similar to (Von Kügelgen et al., 2021), we use linear Logistic Regression Classifiers to measure predictivity ($D_{\text{score}}$). The higher the $D_{\text{score}}$, the more informative is the latent code w.r.t. the gt. SAP score for a gt factor is defined as the difference between the highest and second highest predictivities for it given any latent codes. In our case, SAP(c) can be defined as $|D_{\text{score}}(c|c) - D_{\text{score}}(c|z)|$ and SAP(z) as $|D_{\text{score}}(z|z) - D_{\text{score}}(z|c)|$. The higher the SAP score, higher the separateness of the latent codes. For qualitative evaluations, we performed downstream tasks like generating images through unsupervised content-transformer transfer and latent space visualization. For protein subtomograms, we performed subtomogram averaging Chen et al. (2019) to identify proteins resulting from the latent space clusters.

### 4.1 DUALCONTRAST DISENTANGLES SEVERAL WRITING STYLES FROM MNIST IMAGES

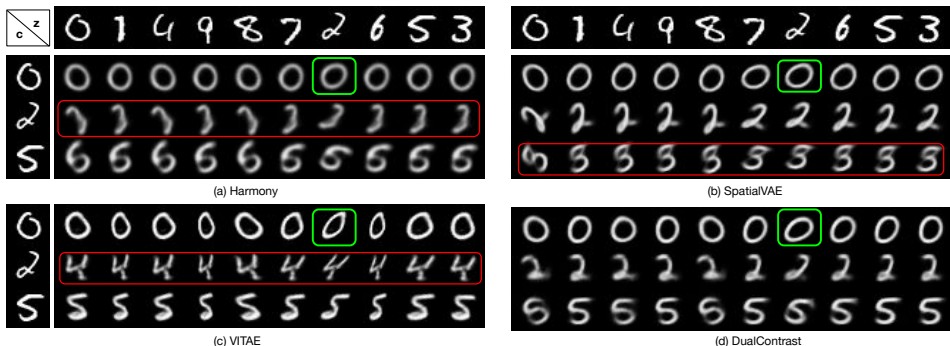

(a) Harmony  (b) SpatialVAE  (c) VITAE  (d) DualContrast

Figure 3: Qualitative Results of Unsupervised **c**-**z** Disentanglement on MNIST obtained by (a) Harmony, (b) SpatialVAE, (c) VITAE, and (d) DualContrast, respectively. Images are generated by the Decoders given content (**c**) code from the leftmost column images and transformation (**z**) code from the topmost row images.

Similar to the related works, we start our experiments with the MNIST dataset of handwritten digits. We performed a quantitative comparison between DualContrast and the baseline approaches, as shown in Table 1. We observed that DualContrast achieves higher $D_{score}(c|c)$ compared to other

explicitly parameterized approaches (Uddin et al., 2022; Bepler et al., 2019; Skafte & Hauberg, 2019).

However, since transformation labels are not present for MNIST, relying entirely on the content prediction performance for disentanglement might be misleading. To this end, we report qualitative results for baseline methods and our approach DualContrast (Fig. 3 and Appendix Fig. 10). In generating images with varying **c** and **z** codes, we observe that Harmony, SpatialVAE, and C-VITAE generate many images with erroneous content and transformations. On the other hand, DualContrast does not make such mistakes. This again suggests better **c**-**z** disentanglement with our approach than existing explicit parameterization approaches.

Moreover, DualContrast disentangles more than in-plane rotation for MNIST. For the digit 0 marked with green box in Fig. 3, Harmony and SpatialVAE simply rotate the image, not capturing the actual writing style of the digit 2 in the top row. VITAE somewhat represented the transformation with its explicit modeling of piecewise linear transformation. On the other hand, DualContrast properly captured the writing style of above digit 2.

We included additional qualitative results, including the **c** and **z** latent space visualization, in the Appendix A.4.

## 4.2 DUALCONTRAST DISENTANGLES VIEWPOINT FROM LINEMOD OBJECT DATASET

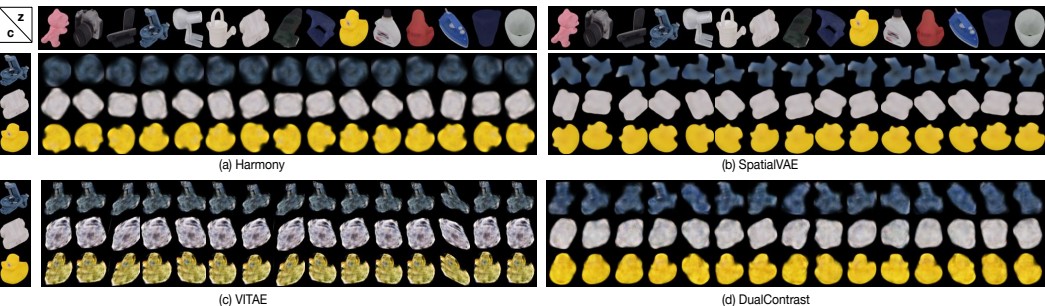

Figure 4: Qualitative Results of Unsupervised **c**-**z** Disentanglement on LineMod obtained by (a) Harmony, (b) SpatialVAE, (c) VITAE, and (d) DualContrast, respectively. Images are generated by the Decoders given **c** code from the leftmost column images and **z** code from the topmost row images. Additional content-transformation transfer results are available on Appendix Fig. 12.

LineMod is an RGB object recognition and 3D pose estimation dataset (Hinterstoisser et al., 2013) with 15 unique object types. The objects are segmented from real-world scenes, and the segmented images visualize the objects from different viewpoints. The entire dataset has $1,313$ images per object category. We used $1,000$ images per category for training and the remaining for testing.

We evaluated the scores $D_{score}$ (Table 1) for DualContrast and baseline approaches, where DualContrast clearly shows the best performance. During evaluation, we used the object identities as ground truth content factors.

We also performed qualitative measurement similar to MNIST and provided the results in Fig. 4 and Appendix 12. We observe that VITAE distorts the images significantly while performing image generation with **c**-**z** transfer. On the other hand, DualContrast does not face such issues. SpatialVAE and Harmony only disentangles in-plane rotation given their design; so changing the **z** code in these methods only rotates the content sample. However, DualContrast can disentangle different viewpoints of the objects as the **z** code and changing **z** changes the viewpoint of objects- reflecting the actual transformation present in the dataset.

## 4.3 DUALCONTRAST DISENTANGLES PROTEIN COMPOSITION FROM CONFORMATIONS IN CRYO-ET SUBTOMOGRAMS AND ENABLES THEIR PRECISE IDENTIFICATION

We created a realistically simulated (Liu et al., 2020) cellular cryo-ET subtomogram dataset of 18,000 samples belonging to 3 different protein identities - Nucleosome, Sars-Cov-2 spike protein, and Fatty Acid Synthase (FAS) unit. We collected 6 different conformations or compositional states

for each protein identity from RCSB PDB (RCSB, 2000). Among them, only nucleosome composition has subtly visible differences across all the 6 states. For each conformation per protein identity, 1,000 subtomograms were generated. Each subtomogram is of size $32^3$ and contains a protein in random orientation and shift with a very low signal-to-noise ratio (Fig. 5). Further details on the dataset properties are provided in the Appendix A.4.

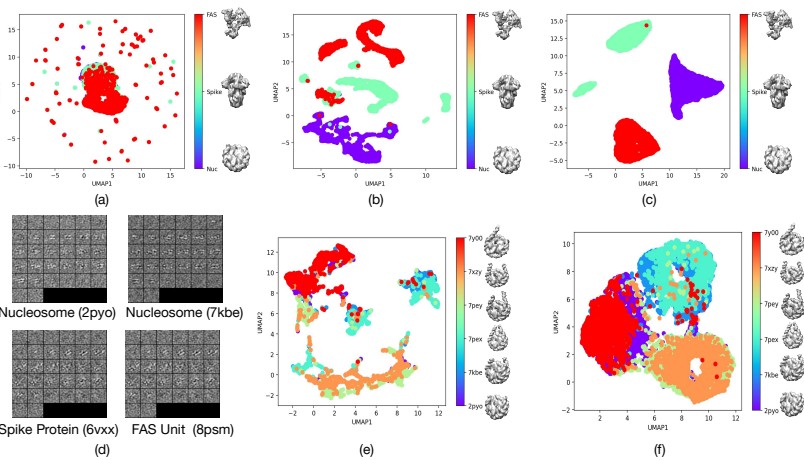

Figure 5: Disentanglement of composition and conformations in cellular subtomogram dataset with slice-by-slice visualization of 4 sample subtomograms. UMAP embedding of **c** codes in (a) Spatial-VAE, (b) Harmony, and (c) DualContrast. (d) Slice-by-slice visualization of x-y slices in 4 sample subtomograms. (e) UMAP embedding of **c** codes in Harmony trained only nucleosome subtomograms. (f) UMAP embedding of **z** codes in Harmony trained with all subtomograms.

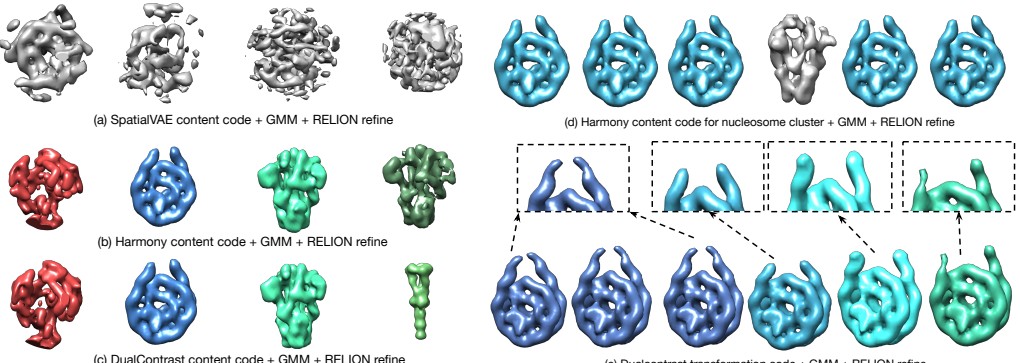

Figure 6: (a,b,c) Structures obtained with RELION refinement for each (4) cluster of subtomograms, whereas the clustering is performed using Gaussian Mixture Modeling (GMM) on **c** codes, predicted by (a) SpatialVAE (b) Harmony, and (c) DualContrast. (d, e) Structures obtained with RELION refinement for the nucleosome subtomograms (identified with previous step) using GMM on (d) Harmony content (**c**) codes, and (e) DualContrast transformation (**z**) codes.

We trained DualContrast and baseline approaches (except VITAE since it could not be trained without designing a 3D CPAB transformation) against the protein subtomogram dataset. We provided the quantitative results in Table 1 and qualitative results in Fig. 5 and Fig. 6 respectively.

For qualitative results, we first perform UMAP visualization of the latent codes obtained by the models. We observed that the UMAP of **c** codes in DualContrast perfectly clustered the protein identities from the dataset, profoundly disentangling the transformations (Fig. 5(c)). We do not see such clustering for Harmony (Fig. 5(b)); rather, protein identities with very different compositions get mixed up due to their conformational variation. SpatialVAE latent code UMAP could not cluster the proteins at all (Fig. 5(a)). We further observe that DualContrast **z** code UMAP clusters have similar nucleosome conformations, showing disentanglement of conformation from protein identity. On the other hand, in Harmony, the transformation factor only represents rotation and translation

by design and can not capture nucleosome conformations. Moreover, even its **c** code for manually selected (not through automatic clustering) nucleosome subtomograms can not show good clustering of conformations (Fig. 5(b)).

To further demonstrate the application of such disentanglement, we perform downstream subtomogram averaging to identify the structures from latent space clusters. Subtomogram averaging obtains readily identifiable high SNR structures from multiple low SNR subtomograms through iterative alignment and averaging Chen et al. (2019). In our experiments, Gaussian Mixture Models (GMM) were applied to content (**c**) codes predicted by models, generating clusters of subtomograms that were averaged using RELION Zivanov et al. (2018) (Fig. 6). SpatialVAE failed to identify proteins, while Harmony identified the FAS, nucleosome, and 2 spike proteins, though 1 spike protein appeared as an unrealistic mixture of FAS and spike protein. DualContrast successfully identified the FAS, nucleosome, and 2 distinct spike protein compositions. For nucleosome conformations, GMM clustering of Harmony's **c** codes for nucleosome cluster subtomograms revealed only 1 nucleosome conformation but mistakenly included a spike protein. In contrast, clustering DualContrast's **z** codes followed by RELION averaging identified 4 distinct nucleosome conformations with subtle structural changes, showcasing its ability to disentangle protein composition and conformation, which is unattained by the other existing methods.

**Ablation Study:** To evaluate the individual contribution of the contrastive losses, we conduct both quantitative and qualitative ablation analyses of DualContrast. We trained (1) DualContrast without any contrastive loss, which is basically a VAE with two latent spaces, (2) DualContrast with only $L = L_{\mathrm{VAE}} + L_{\mathrm{con}\,(z)}$, and (3) $L = L_{\mathrm{VAE}} + L_{\mathrm{con}\,(c)}$. We qualitatively and quantitatively evaluated each model. We show the quantitative results in Table 1 and qualitative results of MNIST in Fig. 7. We provide qualitative results and a detailed discussion on the ablation in the Appendix A.4.4.

(a)

(b)

Figure 7: Content-transformation transfer results from ablation analysis. (a) and (b) shows the results when the model is trained with $L_{\mathrm{VAE}} + L_{\mathrm{con}\,(z)}$ and $L_{\mathrm{VAE}} + L_{\mathrm{con}\,(c)}$ respectively.

## 5 DISCUSSIONS & LIMITATIONS

We introduced an unsupervised content-transformation disentanglement method that, for the first time, does not rely on labels or explicit parameterization of transformations. Our method successfully disentangled transformations that cause subtle pixel-space changes, such as variations in writing styles (MNIST), viewpoint changes (LineMod), and, most importantly, conformational changes in proteins from protein-mixture cryo-ET subtomogram datasets. However, it is not guaranteed to disentangle all transformations, particularly those causing large changes in the pixel space. Instead, the method may classify transformations causing large pixel-space changes as content. Nonetheless, this aligns with scientific imaging, where large changes often reflect compositional shifts and subtle changes represent conformational variations. Disentangling transformations with large pixel-space changes in the pixel space without any explicit parameterization is extremely challenging and might not be practically achievable, but it remains an avenue for future research.

## 6 CONCLUSION

This work focuses on a challenging setting of the unsupervised content-transformation disentanglement problem in scientifically important shape focused image datasets, where the transformation latent code is not explicitly parameterized. To tackle this problem, we propose a novel method termed DualContrast. DualContrast employs generative modeling with a novel contrastive learning strategy that creates positive and negative pairs for content and transformation latent codes. Our extensive experiments show DualContrast's effective disentanglement of challenging transformations across various shape-based image datasets; including simulated cellular subtomograms, where it solved the unexplored problem of isolation of protein conformations from protein identities, as proof of principle.

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

# A  APPENDIX

## OVERVIEW

- Appendix A.1 contains additional discussion on related works.
- Appendix A.2 contains an additional mathematical explanation of DualContrast and Baseline methods.
- Appendix A.3 contains additional details on the experiments.
- Appendix A.4 contains additional results and description on the datasets.

## A.1  DETAILED RELATED WORKS

**Disentangled Representation Learning:** PCA (Halko et al., 2011) and ICA (Hyvärinen & Oja, 2000) can be regarded as very preliminary work in the domain of disentangled representation learning. However, these methods assume linear subspace and do not work well for non-linear high-dimensional datasets. Deep learning-based approaches like Info-GAN (Chen et al., 2016), $\beta$-VAE (Higgins et al., 2016), and their variants (Chen et al., 2018; Kim & Mnih, 2018; Kumar et al., 2018; Kim et al., 2019) have recently gained wide attention as generic approaches for learning disentangled representations. Most of these works manipulated the variational bottleneck to achieve disentanglement of the latent codes. However, these works do not aim toward disentangling any specific factor, *e.g.* , content, group, style, transformation, etc., from the latent codes. Instead, they generate a series of images by traversing through each dimension of the latent space while keeping the remaining dimensions fixed. Thus, they infer the semantic meaning of each dimension of the learned latent factor. Consequently, these methods do not perform well in disentangling any specific generative factor compared to those that aim to disentangle several (two in most cases) specific generative factors (Uddin et al., 2022; Skafte & Hauberg, 2019; Bepler et al., 2019). Unlike these methods, our method specifically disentangles the content and transformation factors of data samples, whereas the content and transformation are defined in Section 3.1.

**cryo-EM Heterogeneous Reconstruction:** There exists several works on single particle cryo-EM and cryo-ET reconstruction, *e.g.* , cryoDRGN2 (Zhong et al., 2021), cryoFIRE (Levy et al., 2022b), cryoAI (Levy et al., 2022a), etc. that performs amortized inference of transformation ($SO(3) \times d^2$) and latent space representing content. However, these works mainly focus on 2D-to-3D reconstruction instead of content-transformation disentanglement. Our work, on the contrary, focuses on content-transformation disentanglement. Though we use reconstruction loss to maximize informativeness of content and transformation factors, our reconstruction is 2D-to-2D or 3D-to-3D, unlike the aforementioned works. Also, our transformation factor is implicit and not explicitly limited to $SO(3) \times d^2$.

**Shape Analysis:** Disentangling content and transformation latent factors have special significance in the domain of shape analysis. Consequently, shape representation learning, modeling, and analysis (Monti et al., 2017; Tan et al., 2018; Palafox et al., 2021; Zhou et al., 2020; Huang et al., 2021; Cosmo et al., 2020) are closely related to our work. Even for shape analysis, PCA can be regarded as one of the primitive methods. Even now, PCA is widely used in the shape analysis of protein complexes (Bakan et al., 2011). Recently, Huang et al. (Huang et al., 2021) demonstrated that PCA with two components on the latent factor learned by an auto-encoder corresponds to content (shape) and style (pose) in 3D human mesh datasets. Nevertheless, PCA is a linear method assuming linear subspaces, which often does not hold true. A line of shape analysis research (Cosmo et al., 2020; Aumentado-Armstrong et al., 2019; Tan et al., 2018; Zhou et al., 2020) has been performed for non-linear disentanglement of content and style factors in 3D mesh or point cloud datasets. The goal of these works is to reduce per-vertex reconstruction loss of 3D meshes or point clouds for content-style-specific generation. These works claim unsupervised disentanglement as they do not require ground truth factors. However, they use the identity information of meshes, which is directly associated with content code. In contrast, our method does not require identity information apriori to learn latent codes specific to shape and code. Moreover, the mentioned works specifically investigate mesh-specific geodesic losses to achieve minimal per-vertex mesh reconstruction. On the other hand, we do not specifically aim to design mesh-specific losses in this work, rather, we propose a generic content-transformation disentanglement approach that can be applied to 3D mesh datasets with necessary modification in the model architecture.

## A.2 METHOD

### A.2.1 CONTENT-TRANSFORMATION DISENTANGLEMENT WITH VARIATIONAL AUTOENCODERS (VAE)

A standard Variational Autoencoder (VAE) presumes data $\mathbf{x}$ to be generated by latent variable $\mathbf{z}$, whereas a standard Gaussian prior is assumed for $\mathbf{z}$.

$$p(\mathbf{x}) = \int p(\mathbf{x}|\mathbf{z})p(\mathbf{z})d\mathbf{z}$$

$$p(\mathbf{z}) = \mathcal{N}(0, \mathbf{I}_d)$$

We extended the standard VAE to a two-latent variable setting. We assume latent variables $\mathbf{z}$ and $\mathbf{c}$ to generate the data $\mathbf{x}$.

$$p(\mathbf{x}) = \iint p(\mathbf{x}|\mathbf{z}, \mathbf{c})p(\mathbf{z})p(\mathbf{c})d\mathbf{c}$$

This setting is similar to VITAE (Skafte & Hauberg, 2019), SpatialVAE (Bepler et al., 2019), and Harmony (Uddin et al., 2022). However, in SpatialVAE (Bepler et al., 2019) and Harmony (Uddin et al., 2022), latent factor $\mathbf{z}$ is restricted as rotation and parameterized transformations, respectively. In SpatialVAE,

$$p(\mathbf{z}) = \text{Unif}(a, b) \tag{4}$$

$$\theta \sim p(\mathbf{z}|\mathbf{x}) \tag{5}$$

$$\mathbf{x}_{\text{cord}} = \mathcal{R}([-1, 1]^{d \times d}; \theta) \tag{6}$$

$$p(\mathbf{x}) = \int p(\mathbf{x}|\mathbf{c}, \mathbf{x}_{\text{cord}})p(\mathbf{c})d\mathbf{c} \tag{7}$$

where $a$ and $b$ are specified constants, $\theta$ are transformation (2D rotation and translation) parameters, $\mathcal{R}$ is the corresponding transformation operator.

On the other hand, in Harmony,

$$\theta = \mathbb{I}(\mathbf{z}|\mathbf{x})$$

$$\mathbf{x}' = \mathcal{T}(x; \theta)$$

$$p(\mathbf{x}') = \int p(\mathbf{x}'|\mathbf{c})p(\mathbf{c})d\mathbf{c}$$

where $\mathbb{I}$ is an identity function, $\theta$ are transformation parameters and $\mathcal{T}$ is the corresponding transformation operator.

Unlike these two methods, in VITAE (Skafte & Hauberg, 2019) and our proposed DualContrast, we use standard Gaussian priors for latent codes $\mathbf{z}$ and $\mathbf{c}$.

$$p(\mathbf{z}) = \mathcal{N}(0, \mathbf{I}_d)$$
$$p(\mathbf{c}) = \mathcal{N}(0, \mathbf{I}_d)$$

However, in VITAE (Skafte & Hauberg, 2019), $\mathbf{z}$ is used to explicitly sample continuous piecewise affine velocity (CPAB) transformation parameter $\theta$, and $\mathbf{c}$ is used to sample appearance samples $\mathbf{x}'$.

$$\theta \sim p(\mathbf{x}|\mathbf{z})$$
$$\mathbf{x}' \sim p(\mathbf{x}|\mathbf{c})$$
$$\mathbf{x} = \mathcal{T}(\mathbf{x}'; \theta)$$

where $\mathcal{T}$ is the transformation operator for CPAB transformation. CPAB transformation parameter is highly expressive compared to affine transformation parameters used in spatialVAE.

Contrary to VITAE (Skafte & Hauberg, 2019), we do not use $\mathbf{z}$ to sample any transformation parameters explicitly; rather use both $\mathbf{z}$ and $\mathbf{c}$ to generate $\mathbf{x}$. To this end, we use a contrastive learning strategy that is described in Section 3 of the main paper. Without explicitly sampling any transformation parameter, we improve the expressiveness of our transformation latent factor $\mathbf{z}$ even more than the CPAB transformation used in VITAE (Skafte & Hauberg, 2019).

### A.2.2   FEATURE SUPPRESSION OF SIMCLR AND MOCO CONTRASTIVE LOSSES:

The contrastive losses used in popular self-supervised learning methods SimCLR (Chen et al., 2020) or MoCo (He et al., 2020) as did not help much in disentangling content and transformation in our experiments. It has been demonstrated that these methods often learn nuisance image features or noise to obtain a shortcut solution to the contrastive objective (Kahana & Hoshen, 2022). This phenomenon is referred to as *feature suppression* of contrastive objectives. We found that using reconstructive loss was necessary to prevent the feature suppression problem.

### A.2.3   CHOICE OF TRANSFORMATION TO CREATE CONTRASTIVE PAIRS

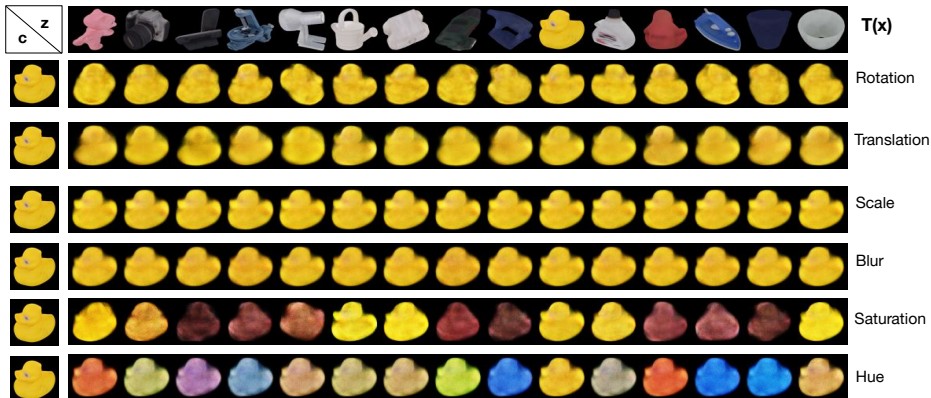

Figure 8: Content-transformation transfer based image generation results using different transformation functions ($T(x)$) to create contrastive pairs. Only rotation ensures transformation factor $z$ to capture object viewpoints- the transformation factor present in the original dataset.

We leveraged different transformation functions $T(x)$ to create contrastive pairs in DualContrast for LineMod RGB object dataset. We used rotation, translation, scaling, blur, saturation, and hue as $T(x)$. We performed both qualitative (Table 2) and quantitative analysis on the effect of different

$T(x)$ for content-transformation disentanglement in DualContrast. We observe that using Scale or Blur makes the transformation factor $\mathbf{z}$ uninformative of the data and it does not capture anything at all. Consequently, changing this $\mathbf{z}$ factor while generating images does not change the image at all for these two codes (Figure 8). On the other hand, using translation shows small negligible differences in the $\mathbf{c}$-$\mathbf{z}$ transfer-based image generation. Color-based transformations like saturation and Hue only change the color of the generated image, instead of affecting its shape-based transformation. Only rotation provides generalization of $\mathbf{z}$ and enables $\mathbf{z}$ to capture viewpoint transformations present in the dataset.

Table 2: Transformation Factors and Corresponding $D_{\text{score}}(c|z)$ and $D_{\text{score}}(c|c)$ values.

| Transformation Factor | $D_{\text{score}}(c|z)$ | $D_{\text{score}}(c|c)$ |
|---|---|---|
| Rotation | 0.48 | 0.95 |
| Translation | 0.65 | 0.98 |
| Scale | 0.52 | 0.91 |
| Contrast | 0.51 | 0.93 |
| Saturation | 0.71 | 0.86 |
| Hue | 0.61 | 0.88 |
| Blur | 0.47 | 0.92 |

## A.3 EXPERIMENTS

### A.3.1 IMPLEMENTATION DETAILS

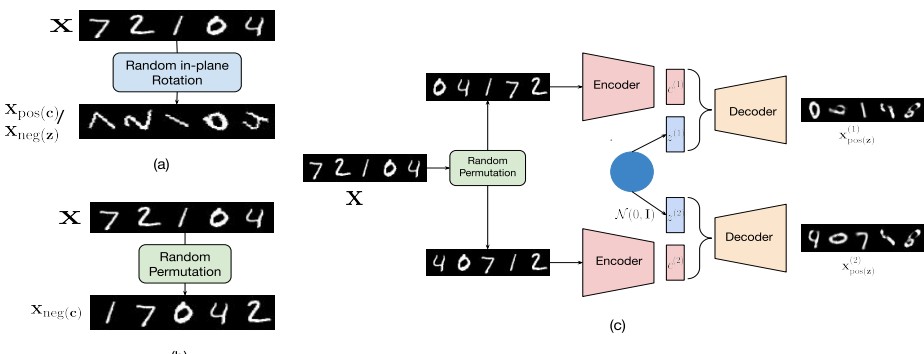

Figure 9: **Visualization of the creation of contrastive pairs for MNIST**. (a) Creating positive pair with respect to (w.r.t) content factor and negative pair w.r.t. transformation. (b) Creating negative pair w.r.t. content. (c) Creating positive pair w.r.t. transformation. We show the processes for a batch of MNIST digits with a batch size of 5.

We implemented our model in Pytorch (version 1.9.0). We used a convolutional neural network (CNN) (3 convolutional layers for MNIST, 4 for others) to implement the encoder and a fully connected network (FCN) (5 layers) to implement the decoder. For subtomograms, we used a 3D convolutional network for the encoder. We do not use any pooling layers in our networks.

While training the models, we use a batch size of 100 and an Adam optimizer with a learning rate of 0.0001. We used a linear learning rate scheduler that decays the learning rate of each parameter group by 0.1 every 50 epochs. We trained our models for 200 epochs. We used the same setting for our models and the baseline models. We used NVIDIA RTX A500 and AMD Radeon GPUs to train the models.

**Choice of latent dimension:** For Harmony (Uddin et al., 2022) and SpatialVAE (Bepler et al., 2019), the transformation latent factor is restricted to dimension 3. For VITAE (Skafte & Hauberg, 2019), SimCLR (Von Kügelgen et al., 2021), and our DualContrast, there is no such restriction on the transformation latent factor dimension and same dimension was used as the content factor. For

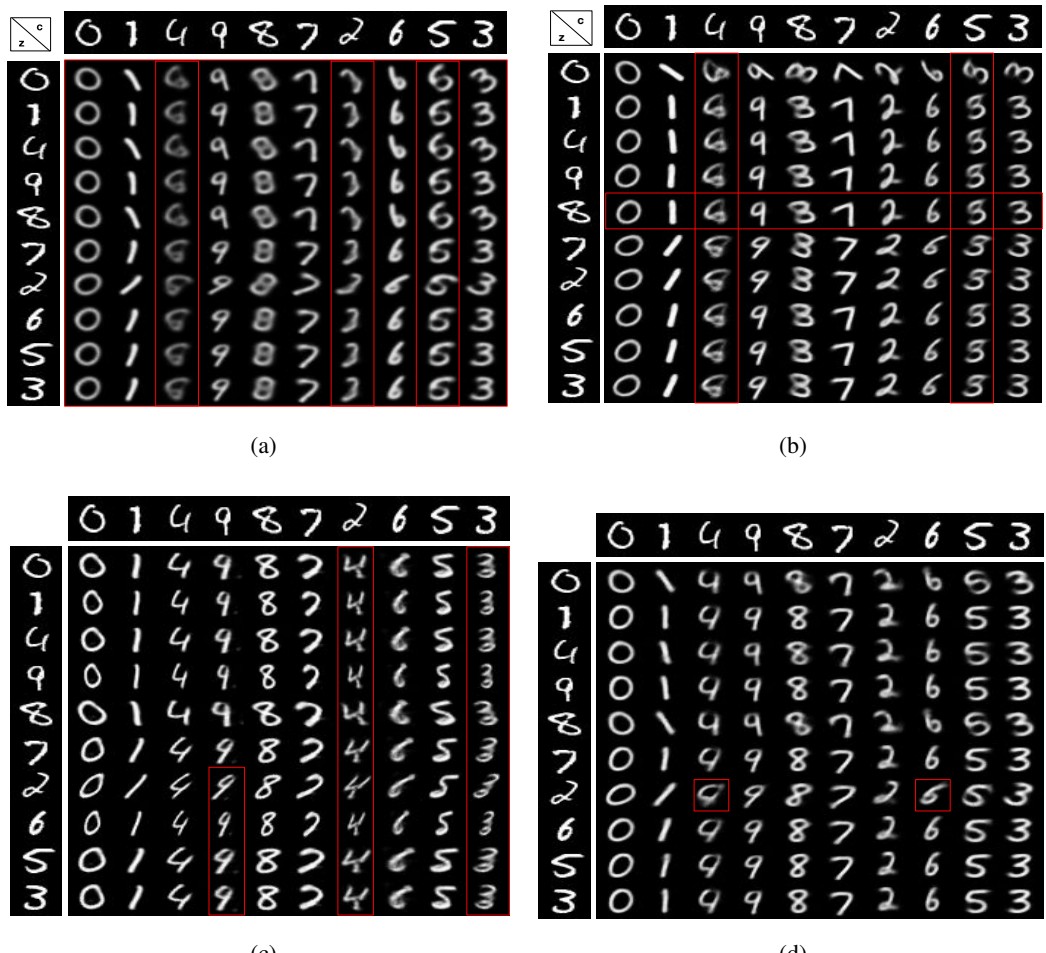

Figure 10: Content-transformation transfer results from ablation analysis (a), (b), (c), and (d) shows the results with Harmony (Uddin et al., 2022), Spatial-VAE (Bepler et al., 2019), VITAE (Skafte & Hauberg, 2019), and DualContrast respectively. When generating image grids, the transformation factor is uniform across rows, and the content factor is uniform across columns. Erroneous generations (both in terms of content and transformation) are marked within red boxes.

all the methods, the dimension of the content factor was set as 10 for MNIST, 50 for subtomogram dataset. For hyperparameters $\gamma_c$ and $\gamma_z$, we set a small value ($\approx 0.01$) in our experiments. The values determine how strictly the content factor and the transformation factor should mimic the prior standard multivariate gaussian distribution.

## A.4 ADDITIONAL RESULTS

### A.4.1 MNIST

We use the commonly used MNIST dataset to initialize our experiments. MNIST is a dataset in the public domain that the research community has extensively used. It contains grayscale images of handwritten digits. Each image is of size $28 \times 28$. The training set contains 60,000 images, whereas the test set contains 10,000. We use the same train test split for our experiments. We show a full visualization of content-transformation transfer based image generation of Figure 3 in Figure 10. We also include tSNE embedding of the content codes inferred by the models on the MNIST test dataset associated with class labels (Figure 11). On the embedding space, DualContrast clearly shows superior clustering performance.

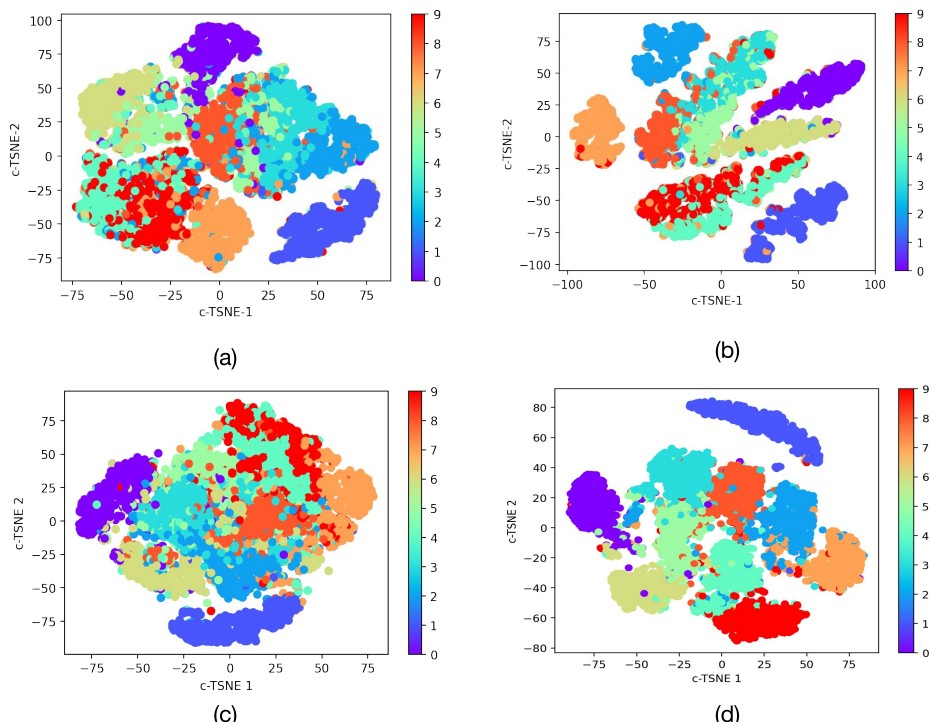

Figure 11: tSNE embedding plots of content latent factor learned by the unsupervised content-transformation disentanglement methods. (a), (b), (c), and (d) shows the results for Harmony (Uddin et al., 2022), Spatial-VAE (Bepler et al., 2019), C-VITAE (Skafte & Hauberg, 2019), and DualContrast respectively. Overall, DualContrast shows superior performance.

### A.4.2 LINEMOD

LineMod (Hinterstoisser et al., 2013) dataset is originally designed for object recognition and 6D pose estimation. It contains 15 unique objects: 'ape', 'bench vise', 'bowl', 'cam', 'can', 'cat', 'cup', 'driller', 'duck', 'eggbox', 'glue', 'hole puncher', 'iron', 'lamp' and 'phone', photographed in a highly cluttered environment. We use a synthetic version of the dataset (Wohlhart & Lepetit, 2015), which has the same objects rendered under different viewpoints. The dataset is publicly available at this url. The dataset is publicly available under MIT License.

This dataset has $1,313$ images per object category. We used $1,000$ images per category for training and used the remaining for testing. For many objects, the object region covers only a tiny part of the original image. To this end, we cropped the object region from the original image using the segmentation masks provided with the original dataset. After cropping the object regions, we padded 8 pixels to each side of the cropped image and then reshaped the padded image to the size of $(64, 64, 3)$. Thus, we prepared the training and testing datasets for content-transformation disentanglement in LineMod. We used the same dataset and train-test splits for our model and the baseline models. The associated processing codes are provided in the supplementary material.

We trained our proposed DualContrast, VITAE (Skafte & Hauberg, 2019), SpatialVAE (Bepler et al., 2019), and Harmony (Uddin et al., 2022) on the LineMod dataset. We provide qualitative results of image generation with content-transformation transfer in Fig. 12 obtained with each method. It is noticeable that both Harmony (Uddin et al., 2022) and SpatialVAE (Bepler et al., 2019) have shown good performance when it comes to reconstruction. However, these two methods can only perform rotation and translation of the objects with explicitly defined transformations and can not capture complex transformations, e.g., projection, viewpoint change, etc. Compared to SpatialVAE and Harmony, VITAE (Skafte & Hauberg, 2019) can perform better transformation transfer but performs poor reconstruction. Nevertheless, DualContrast stands out for its superior ability to perform transformation transfer while ensuring optimal performance in reconstruction.

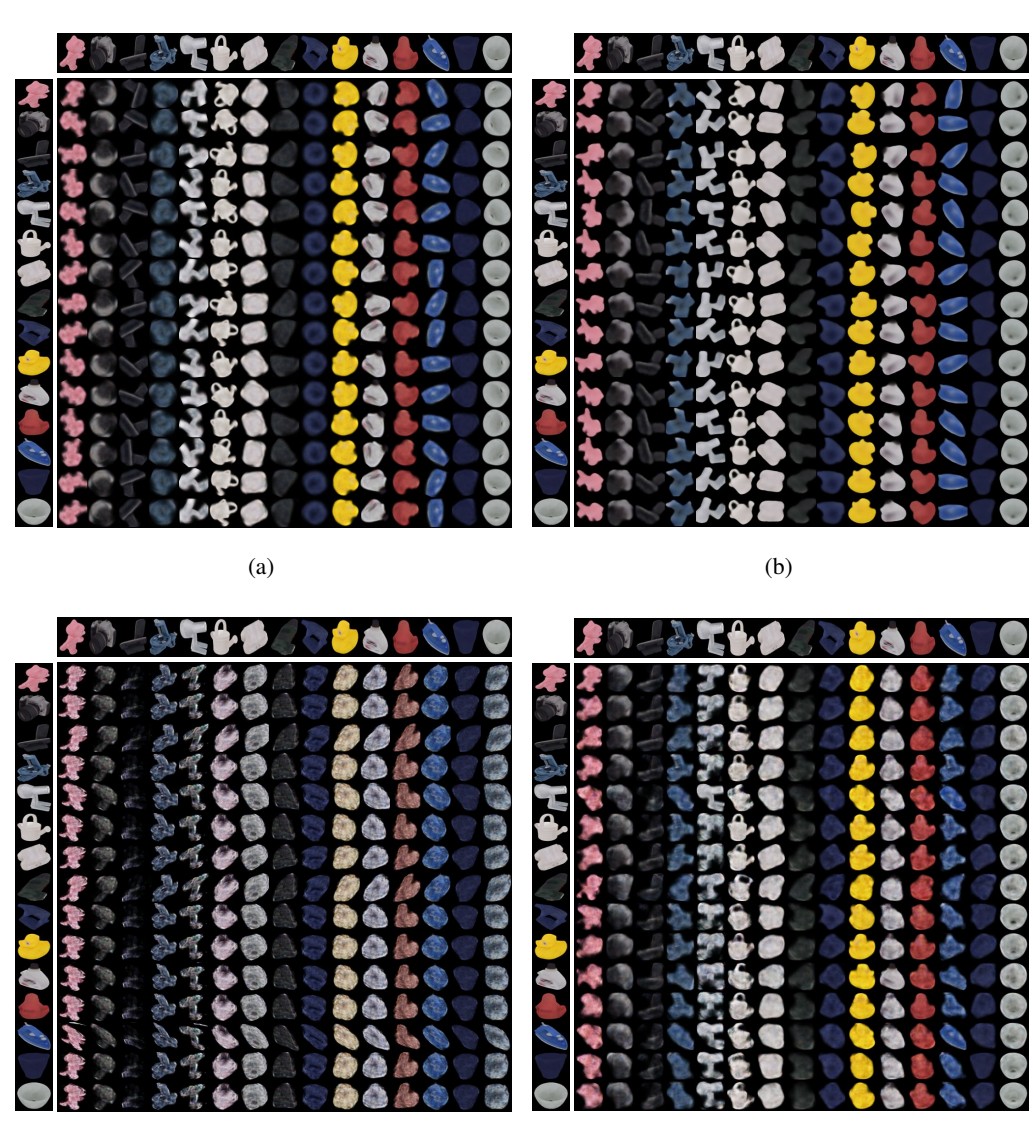

(a)

(b)

(c)

(d)

Figure 12: Qualitative results of image generation with content-transformation transfer obtained by (a) Harmony (Uddin et al., 2022), (b) SpatialVAE (Bepler et al., 2019), (c) VITAE (Skafte & Hauberg, 2019), and (d) DualContrast respectively. Harmony and SpatialVAE perform well in reconstruction. but can only perform rotation and translation with its explicitly defined transformations. On the other hand, VITAE can comparatively perform better disentanglement with very poor reconstruction results, distorting the images. On the other hand, DualContrast provides superior content-transformation transfer with optimal performance in reconstruction.

### A.4.3 PROTEIN SUBTOMOGRAM DATASET

We created a realistic simulated cryo-ET subtomogram dataset of 18,000 subtomograms of size $32^3$. The dataset consists of 3 protein classes of similar sizes- Nucleosomes, Sars-Cov-2 spike protein, and Fatty Acid Synthase Unit. These proteins are significantly different in terms of their composition, which determines their different identities. Moreover, structures for all of these three types of proteins have been resolved in cellular cryo-ET (Klein et al., 2020; Harastani et al., 2022; de Teresa-Trueba et al., 2023), which makes it feasible to use them for our study. Moreover, cellular

cryo-ET is the primary method to capture these proteins inside the cells in their native state (Doerr, 2017).

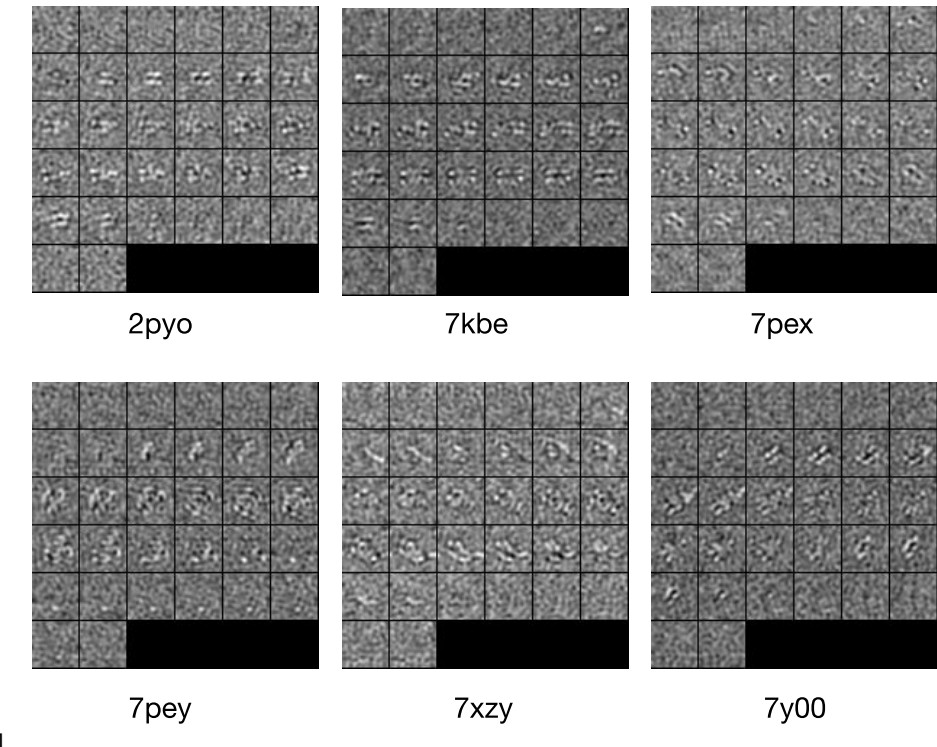

2pyo        7kbe        7pex

7pey        7xzy        7y00

]

Figure 13: 3D slice-by-slice visualization of Nucleosome subtomograms. Each subtomogram is associated with the PDB ID of the original structure.

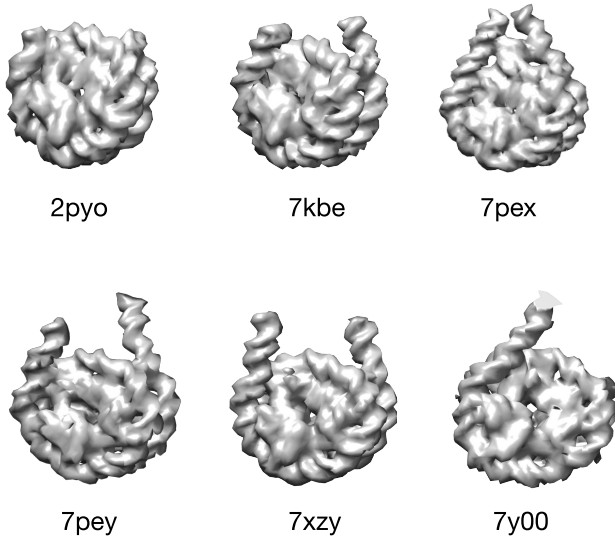

2pyo        7kbe        7pex

7pey        7xzy        7y00

Figure 14: Isosurface visualization of Nucleosome Density Maps. Each density map slightly varies in terms of conformation. They are associated with the PDB IDs in the figure.

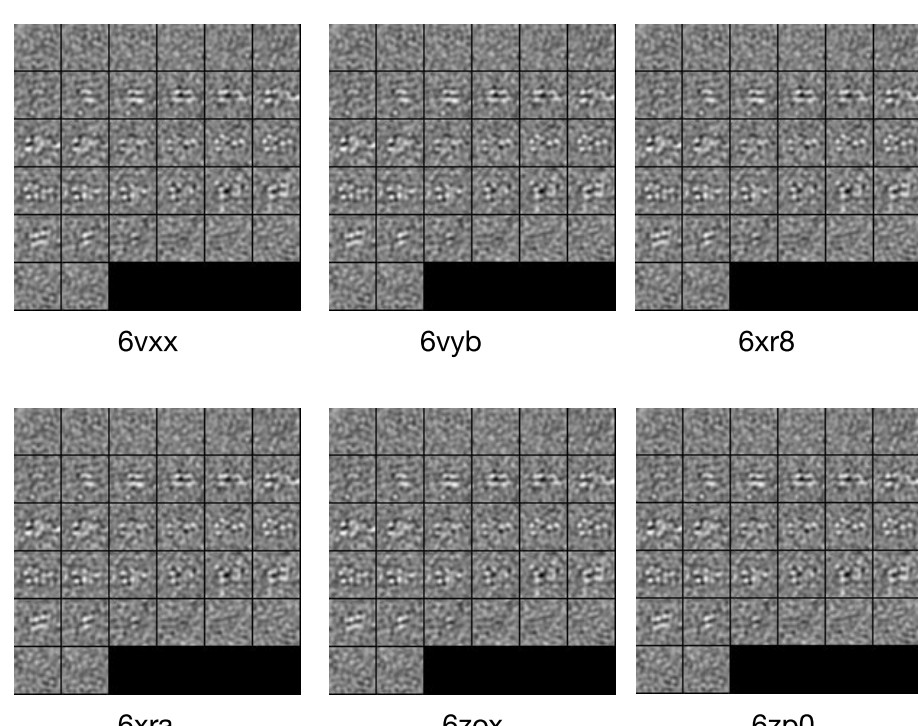

Figure 15: 3D slice-by-slice visualization of Spike Protein Subtomograms. Each subtomogram is associated with the PDB ID of the original structure.

For each protein class, we collected 6 different protein structures from the RCSB PDB website (RCSB, 2000). RCSB PDB is a web server containing the structure of millions of proteins. For nucleosomes, we collected PDB IDs '2pyo', '7kbe', '7pex', '7pey', '7xzy', and '7y00'. All of these are different conformations of nucleosomes that slightly vary in composition (Figure 14). For sars-cov-2 spike proteins, we collected PDB IDs '6vxx', '6vyb', '6xr8', '6xra', '6zox', and '6zp0'. Among them, only '6xra' shows much variation in structure from the other ones, and the rest of the PDB IDs are very similar in structure. For Fatty Acid Synthase (FAS) Unit, we used PDB IDs '8prv', '8ps1', '8ps9', '8psj', '8psm', '8psp'. They also vary very slightly in terms of the structure.

After collecting these 18 PDB structures as PDB files, we used EMAN PDB2MRC (Tang et al., 2007) to create density maps (as MRC file extension) from the PDB files. We create density maps of size $32^3$ with 1 nm resolution. We then randomly rotate and translate each density map and create 1000 such copies. We then convolve the density maps with CTF with CTF parameters common in experimental datasets (Defocus -5 nm, Spherical Abberation 1.7, Voltage 300 kV). Afterward, we add noise to the convolved density maps so that the SNR is close to 0.1. Thus, we prepare 18,000 realistic subtomograms with 3 different protein identities, each with 6 different conformations. We uploaded the entire dataset anonymously at `https://zenodo.org/records/11244440` under CC-BY-SA license. Sample subtomograms for nucleosomes, spike proteins, and FAS units are provided in Figure 13, Figure 15, and Figure 16 respectively. The figures show 3D slice-by-slice visualization for each conformation of the corresponding protein.

We could not train VITAE (Skafte & Hauberg, 2019) on subtomogram datasets since it did not define any transformation for 3D data. Designing CPAB transformation for 3D data by ourselves was challenging. However, we trained SpatialVAE and Harmony as baselines against our subtomogram dataset. Between these two, spatialVAE could not distinguish the protein identities with high heterogeneity at all, which is evident by its embedding space UMAP (Figure 5). Only Harmony and DualContrast showed plausible result, where DualContrast showing much superior disentanglement than Harmony (Figure 5).

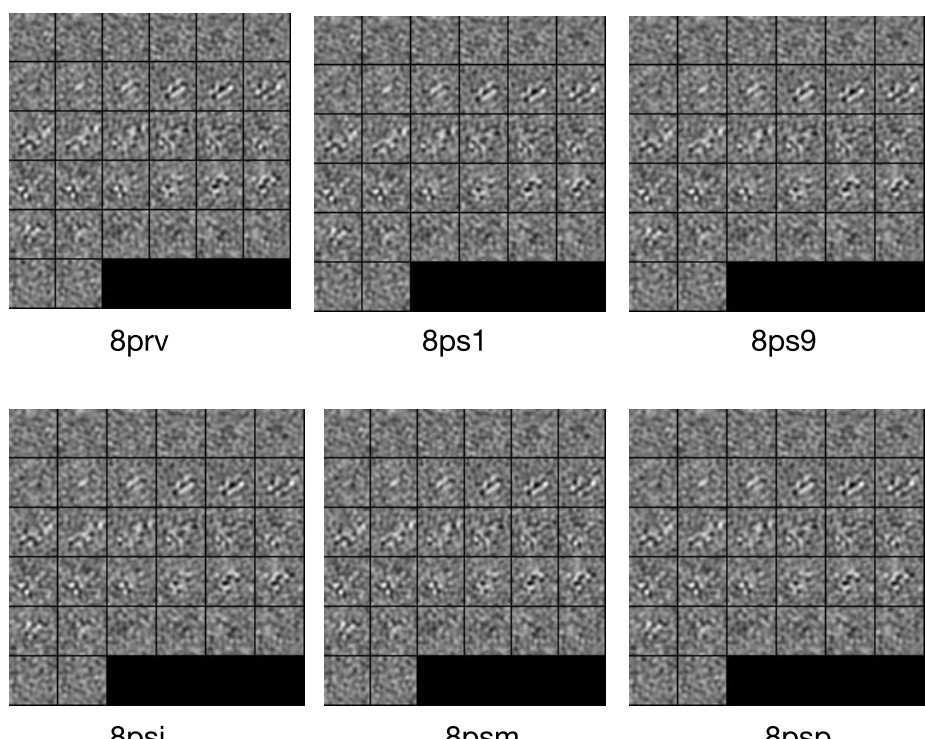

Figure 16: 3D slice-by-slice visualization of FAS subtomograms. Each subtomogram is associated with the PDB ID of the original structure.

### A.4.4    ABLATION STUDY

To evaluate the individual contribution of the contrastive losses, we conduct both quantitative and qualitative ablation analyses of DualContrast. We trained (1) DualContrast without any contrastive loss, which is basically a VAE with two latent spaces, (2) DualContrast with only $L = L_{\text{VAE}} + L_{\text{con (z)}}$, and (3) $L = L_{\text{VAE}} + L_{\text{con (c)}}$. We qualitatively and quantitatively evaluated each model.

For (1), the $D_{\text{score}}$ is almost the same for both **c** and **z** codes, indicating equal predictivity of the digit classes by both codes. This is obvious given that the model has no inductive bias to make different codes capture different information. In model (2), using contrastive loss w.r.t. only **z** factor makes it uninformative of the data. Thus, it provides a small $D(c|z)$ score as desired, but the changing **z** does not affect the image generation (Fig. 7). On the other hand, in the model (3), using contrastive loss w.r.t. only **c** gives a less informative **c** factor, a lower $D(c|c)$ score, and a higher $D(c|z)$ score, contrary to what is desired. These results indicate that contrastive loss w.r.t to both codes is crucial for the desired disentanglement.

Furthermore, we investigated whether using only positive pairs or negative pairs for both codes is sufficient for disentanglement. Nevertheless, we found that both leads to suboptimal disentanglement. If only negative pairs are used, only rotation is disentangled. If only positive pairs are used, then the transformation code becomes uninformative of the data, similar to the degenerate solution.

We provide further quantitative results on the ablation study in Figure 17. The image grids show decoder-generated images where the content factor is used from the corresponding topmost row, and the transformation factor is used from the corresponding leftmost column image.

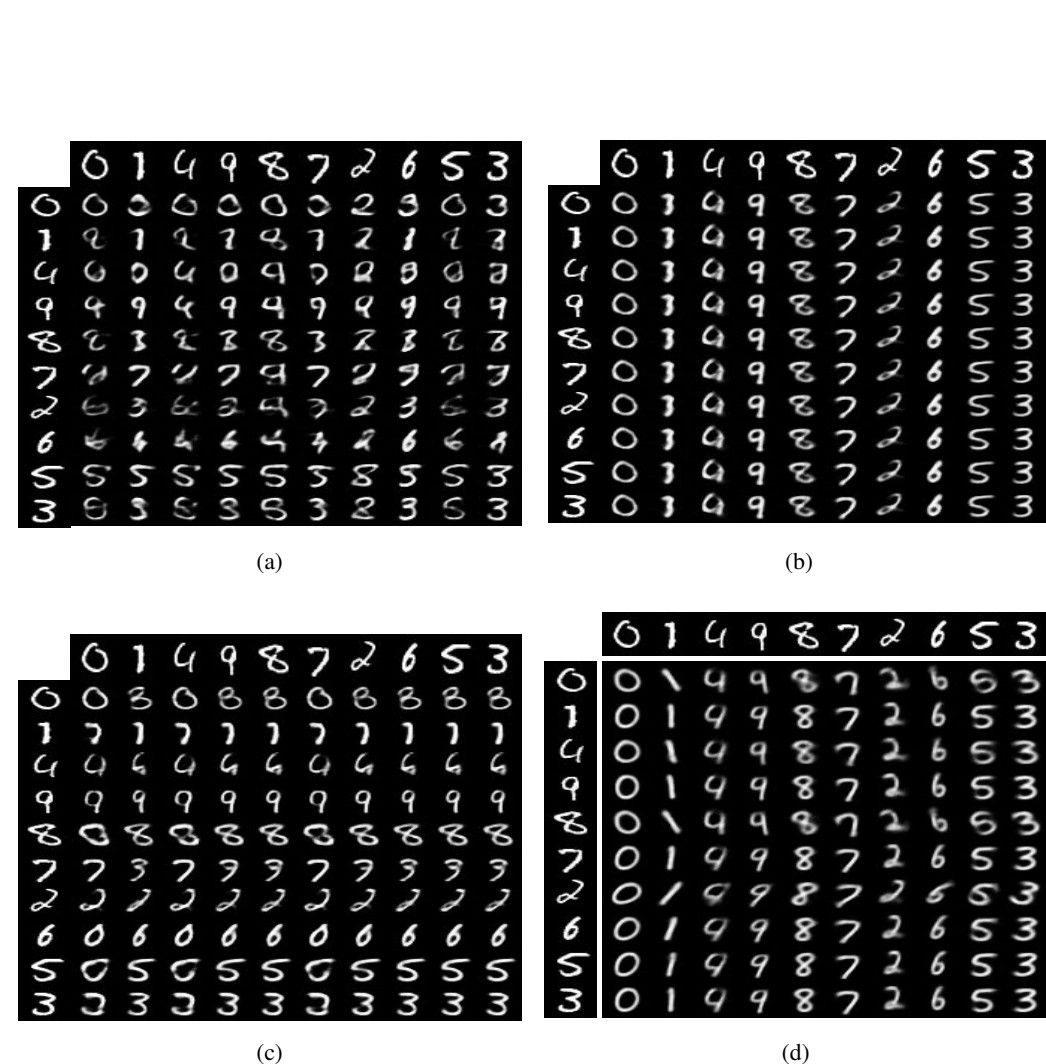

Figure 17: Content-transformation transfer results from ablation analysis. (a), (b), (c), and (d) show the results when the model is trained with $L_{\text{VAE}}$, $L_{\text{VAE}} + L_{\text{con (z)}}$, $L_{\text{VAE}} + L_{\text{con (c)}}$ and full objective respectively.

