# OpenReview forum: "DualContrast: Unsupervised Disentangling of Content and Transformations with Implicit Parameterization"
_ICLR.cc/2025/Conference — Submitted to ICLR 2025_

### Official Review · Reviewer_aQPG · 2024-10-28

**Soundness:** 2
**Presentation:** 3
**Contribution:** 2
**Rating:** 3
**Confidence:** 3

**Summary:**

In this paper, the authors propose a model that learn unsupervised representation for  "shape-focused images". In particular, their method, DualContrast, learn to disentangle "content" and "transformation" in an unsupervised fashion. The model is trained with an a combination of 2 contrastive losses (one for context, one for transformations) and a VAE loss (the VAE is used to sample positive samples for the CL of transformations). The authors show results of the proposed model on multiple small/toysh datasets.

**Strengths:**

+ The paper is well written and easy to follow
+ The idea of disentangling features is an important problem in many applications of machine learning
+ The proposed approach is simple and well motivated

**Weaknesses:**

-  The author often mention that the work focus on "shape-focused real-world images", but they only applied in very simplified, toysh settings, very far from "real-world images". Even the CryoEM task is a very simplified task.
- The choice of positive/negative samples for each factor is very ad-hoc. The paper lacks explanation and empirical validation on why this choice makes sense versus others.
- I found very strange the choice of using VAE generated samples as data to train the contrastive loss. This idea of using generated smples to train a model is not well understood. This approach might have worked in the very simplified tasks tested on the paper, but It is very unlikely that the proposed model would work on any real-world dataset.
- I also find the experimental results a bit weak. First, the datasets utilized in this work are very simple and results on them probably wont guarantee their utility on real-world problems. Second, the metrics utilized on Table 1 are not particularly significant. Third, most of the results are qualitative and based on one or two images that can potentially be cherry picked.

**Questions:**

- On L147, the authors say that they model "is highly effective in disentangling wide range of transformations from the content in various shape-focused image datasets by only using simple rotation for creating contrastive pairs since the representation for disentangled rotation generalizes over other shape transformations." Could they elaborate on this? Why is it the case? Where is it shown on the paper? How can we be sure it will work to other modalities besides the toy tasks tested?
- Is the VAE trained at the smae time as the contrastive losses? Since the VAE is used to generated samples for the CLs, how do training jointly vs training in two stages (VAE followed by CLs) change the performance?

---

> ### Comment · Reviewer_aQPG · 2024-11-28
>
> I thank the authors for their rebuttal. However, the rebuttal did not address most of my initial concerns (related to model design and empirical evaluation). I would suggest the authors to really improve experiments and evaluation metrics (eg, better metrics, better comparison to SOTA, more relevant datasets) and re-submit to a next conference. I keep my rating.

---

> ### Author Response · Authors · 2024-11-28
> **Individual Response to Reviewer aQPG (Part 1)**
>
> Dear Reviewer aQPG,
>
> Please look into the individual responses to your initial comments to find out if it addresses your concern. **We did not clarify all your concerns in the global response since the global response only reflects the major changes in the manuscript, not your individual concerns.**
>
> > “The author often mention that the work focus on "shape-focused real-world images", but they only applied in very simplified, toysh settings, very far from "real-world images". Even the CryoEM task is a very simplified task.”
>
> We think you are confusing the real world with ‘natural images’. The shape-focused scientific images are also ‘real-world’; however, they are far from natural images found in ImageNet. Our revised manuscript mostly replaced ‘scientific images’ with ‘real-world’ to remove the confusion. In most shape-focused scientific image datasets, the transformations include the subtle changes in the pixel space, which we aimed to disentangle from the content. And as you mentioned cryoEM as a simplified task, I think you are confusing cryoEM with cryoET. CryoET contains 3D images of protein complexes with different identities and conformations, whereas cryoEM contains 2D images of protein complexes with the same identity with the same or different conformations. So, for cryoEM, disentangling, conformation, and identity is simpler, but not for cryoET.
>
> > “The choice of positive/negative samples for each factor is very ad-hoc. The paper lacks explanation and empirical validation on why this choice makes sense versus others.”
>
> The problem we are targeting is specific to unsupervised content and transformation disentanglement without explicit parameterization of the transformation. Consequently, the choice of positive/negative samples for the content and transformation factors is also specific to these factors. In the ablation study (L517-L529) in the main manuscript and Appendix Sec A4.4, we empirically validated this choice. We have mentioned that using contrastive factors for only one factor optimizes the reconstruction through the other factor, making the latter factor capture all the information of the data (degenerate solution). We have mentioned whether using only positive or negative pairs for both codes is sufficient for disentanglement. We found that both leads to suboptimal disentanglement. If only negative pairs are used, only rotation is disentangled. If only positive pairs are used, then the transformation code becomes uninformative of the data, similar to the degenerate solution. We mentioned that we did not use the contrastive losses used in SimCLR and MoCo since they resulted in feature suppression (see Appendix Section A2.2). We demonstrated empirical evidence on why we chose rotation (Appendix Section A2.3, Figure 8, Table 2, L286 in the main manuscript.
>
> Do you have any alternate choices in mind? If so, please mention them so we can have a logical discussion about whether the alternate choice could have been used.
>
>
> > “I found very strange the choice of using VAE generated samples as data to train the contrastive loss. This idea of using generated samples to train a model is not well understood. This approach might have worked in the very simplified tasks tested on the paper, but It is very unlikely that the proposed model would work on any real-world dataset.”
>
> We use a bi-latent space VAE. We use one latent space for content and another for transformation. To create positive contrastive pairs for transformation, we use the generated samples from the transformation latent space using a prior Gaussian distribution. Since the reconstruction and contrastive losses are used simultaneously, the losses concerning other contrastive pairs make the data samples with subtle conformational changes closer to the transformation latent space. When we sample them to create positive contrastive pairs and encourage their similarity in the latent space, the samples with subtle conformational changes become closer in the transformation latent space. Thus, transformation captures the subtle pixel space changes and the in-plane rotation in the data. However, this phenomenon was not observed when we used other than in-plane rotation to create contrastive pairs. We regard this as an empirical finding of our manuscript.
>
> Again, we respectfully disagree that the tasks in this paper were very simplified. The cryo-ET scenario was not simple, given multiple levels of heterogeneity, pose and shift confounding, and high noise in the dataset. Given the results on MNIST, LineMod, and cryo-ET subtomograms, the proposed method is likely to work similarly and disentangle transformations that cause small pixel-space changes from scientific image datasets. However, as mentioned earlier, real-world natural images are very different, and our study does not consider those images.

---

> ### Author Response · Authors · 2024-11-28
> **Individual Response to Reviewer aQPG (Part 2)**
>
> > “I also find the experimental results a bit weak. First, the datasets utilized in this work are very simple and results on them probably won’t guarantee their utility on real-world problems. Second, the metrics utilized on Table 1 are not particularly significant. Third, most of the results are qualitative and based on one or two images that can potentially be cherry picked.”
>
> We want to iterate again that the problem does not concern real-world natural images, where disentangling transformations from content is not very useful. The datasets utilized in this work, particularly the cryo-ET dataset, perfectly fit the use case. For Table 1, we added metrics and results for the cryo-ET datasets. We also disagree that most results are based on one or two images. Due to the page limit, the unsupervised content-transformation transfer results for MNIST and LineMod in the main manuscript contain fewer (30-45) images; we provided similar results with more images in Appendix (Fig. 10 and Fig. 12). Also, the latent space visualizations (Fig. 5, Appendix Fig. 11) and the downstream subtomogram averaging (Fig. 6) uses information from all the images in the datasets.
>
> The answers to your questions are below:
>
> >”On L147, the authors say that they model "is highly effective in disentangling wide range of transformations from the content in various shape-focused image datasets by only using simple rotation for creating contrastive pairs since the representation for disentangled rotation generalizes over other shape transformations." Could they elaborate on this? Why is it the case? Where is it shown on the paper? How can we be sure it will work to other modalities besides the toy tasks tested?”
>
> We have moderated the statement in the contributions section of the Introduction (L143-L144). We have demonstrated the disentanglement of these transformations throughout the results section of our paper (Table 1, Fig. 3-6, Fig. 10-12). As for other modalities, I think we mentioned this in previous answers and the Discussions & Limitations section.
>
> >”Is the VAE trained at the smae time as the contrastive losses? Since the VAE is used to generated samples for the CLs, how do training jointly vs training in two stages (VAE followed by CLs) change the performance?
>
> Yes, the VAE is trained at the same time as the contrastive losses. We mentioned this in our method section and in the caption for Figure 2. We have tried VAE followed by CLs; however, that did not result in any disentanglement (SAP score was close to 0.0).

---

> ### Author Response · Authors · 2024-11-28
> **Response to Official Comment by Reviewer aQPG**
>
> Please look into the individual responses to see if they address your concerns or confusion. Even if you are determined to stick to your initial rating, we respect your judgment. Nevertheless, we would very much appreciate if you could be specific about the "better metrics", "better comparison", and "more datasets" you are referring to.
>
> Moreover, the paper was submitted to the primary area of "applications to physical sciences (physics, chemistry, biology, etc.)." We would greatly appreciate it if you took this issue into account for your final judgment.

---

> ### Comment · Reviewer_aQPG · 2024-12-01
>
> I acknowledge and appreciate the authors' further responses. After reading the rebuttal, I will keep my score, as many of my concerns are not entirely addressed.
>
> Indeed I meant CryoET when I wrote CryoEM on my initial post. Independently, CryoEM/CryoET/related technologies are vast research domains with many important applications in biology. What I meant on my initial post is that the experiments conducted on the paper on this application are simple/toy tasks that does not reflect real use-cases of those technologies. Moreover, the main takeaway result on CryoET experiments was a few UMAP plots (Fig 5) or a few qualitative figures (Fig 6).
>
> By "better datasets" I mean something that is not MNIST/LineMod/StarMen. If the focus would be on computer vision applications, the the authors should focus on dataset that are relevant to that community (those are not). If the focus is "applications to physical sciences", maybe the paper could really focus on that topic instead of mostly very toy computer vision datasets. For example, the authors could show results on more than only 3 protein domains, perhaps use experimental data instead of simulated ones (if that is possible), or find some other dataset related to physical sciences to experiment with.
>
> By "better metrics/comparisons", I mean not only show one UMAP figure and one qualitative illustration as validation of the proposed method---that is simply not enough. I am not an specialist on CryoET, but I am sure the community has many metrics to evaluate quality of their approaches. The D_score is a good start, but that metric alone does not really validates the method in any way.

---

> ### Author Response · Authors · 2024-12-02
> **Regarding misunderstandings of the work (Response to reviewer aQPG): Part 1**
>
> Thanks for your detailed response and suggestions. We very much appreciate your time. Nevertheless, we think you have a few misunderstandings regarding our work, contribution, and qualitative evaluation. We have clarified them below:
>
>
> **Type of contribution- Fundamental vs Incremental:** We view our work as a fundamental contribution rather than an incremental one. In cryo-EM/ET, many incremental works focus on tasks like particle picking or determining the location of protein complexes from raw images, aiming to improve state-of-the-art performance. In contrast, our work addresses a novel task: disentangling content and transformations where the transformations, such as conformational changes, lack a well-defined parametric form.
> This is similar to works like SpatialVAE [1], which introduced the task of disentangling content from 2D rotations and translations, and Harmony [2], which extended this to disentangling content from parameterized transformations, including 2D and 3D affine transformations. By addressing this new challenge, our work expands the boundaries of what is achievable in the field.
>
> **Takeaway from cryo-ET results:** The main takeaway from the cryo-ET part of this paper is the demonstration that protein complexes with varying compositions and conformations can be identified from a collection of cryo-ET subtomograms in an unsupervised manner- a capability that previous methods lacked. Earlier approaches were limited to either identifying a few distinct protein complexes from a collection of images or analyzing a few conformations of a single protein complex within a dedicated dataset. However, when presented with collections containing multiple distinct protein complexes and multiple conformations, these methods could, at best, identify only a subset of the complexes and failed entirely to distinguish their conformations. This limitation is clearly illustrated in Fig. 6.
>
> **Qualitative Evaluation:** Regarding the qualitative evaluation, the only way to identify the presence of a protein complex in a collection of subtomogram images is to perform clustering, then subtomogram averaging, and observe the subtomogram averaging result. This is precisely what we have done (in Fig 6). If you look into the relevant works [3-8] in cryo-EM and cryo-ET domains, they have all done the same.
>
> Moreover, the evaluation was performed using the entire dataset; we inferred the latent codes for all the images in the dataset, performed GMM clustering, and then performed subtomogram averaging for each cluster. Fig. 6 shows the subtomogram averaging results. Fig. 5 shows the UMAP of the latent codes for all the images in the dataset. We did not cherry-pick a few images and show them, which would also be meaningless in this scenario.
>
> **Quantitative Evaluation:** One possible quantitative evaluation is to demonstrate how predictive each latent code is for the ground truth factor and to what extent they are separate. In the revised version, we have performed these quantitative evaluations for our cryo-ET dataset (see Table 1, blue colored part) with D_score and SAP score respectively.
>
> **Use of MNIST and LineMod:** We set out to disentangle the composition (semantic content) and conformation (transformations causing subtle voxel-level changes) of protein complexes in cryo-ET subtomogram datasets by framing the problem as an unsupervised content-transformation disentanglement task. These datasets present protein complexes in diverse 3D poses, with varying compositions and conformations, making the task inherently complex.
> To address this, we adopted a strategy of starting with simpler datasets, such as MNIST and LineMod (please note starmen was excluded in the revised paper), which share similar problem contexts. This strategy is consistent with prior works. For instance, SpatialVAE [1] and Harmony [2], while targeting content-transformation disentanglement in cryo-EM and cryo-ET, first validated their methods on MNIST. Similarly, NAISR [9], designed for interpretable shape analysis in medical imaging, initially tested its approach on the starmen dataset to establish its effectiveness on simpler cases. Following this strategy, we demonstrated our method’s success on MNIST and LineMod before applying it to cryo-ET subtomograms for the final experiments.
>
> (References in Part 2)

---

> ### Author Response · Authors · 2024-12-02
> **Regarding misunderstandings of the work (Response to reviewer aQPG): Part 2**
>
> **References**
>
> [1] Bepler, Tristan, et al. "Explicitly disentangling image content from translation and rotation with spatial-VAE." Advances in Neural Information Processing Systems 32 (2019).
>
> [2] Uddin, Mostofa Rafid, et al. "Harmony: a generic unsupervised approach for disentangling semantic content from parameterized transformations." Proceedings of the IEEE/CVF Conference on Computer Vision and Pattern Recognition. 2022.
>
> [3] Levy, Axel, et al. "Amortized inference for heterogeneous reconstruction in cryo-em." Advances in neural information processing systems 35 (2022): 13038-13049.
>
> [4] Zeng, Xiangrui, et al. "High-throughput cryo-ET structural pattern mining by unsupervised deep iterative subtomogram clustering." Proceedings of the National Academy of Sciences 120.15 (2023): e2213149120.
>
> [5] Zhong, Ellen D., et al. "Reconstructing continuous distributions of 3D protein structure from cryo-EM images." International Conference on Learning Representations.
>
> [6] Zhong, Ellen D., et al. "CryoDRGN: reconstruction of heterogeneous cryo-EM structures using neural networks." Nature methods 18.2 (2021): 176-185.
>
> [7] Powell, Barrett M., and Joseph H. Davis. "Learning structural heterogeneity from cryo-electron sub-tomograms with tomoDRGN." Nature Methods (2024): 1-12.
>
> [8] Zhong, Ellen D., et al. "Cryodrgn2: Ab initio neural reconstruction of 3d protein structures from real cryo-em images." Proceedings of the IEEE/CVF International Conference on Computer Vision. 2021.
>
> [9] Jiao, Yining, et al. "$\texttt {NAISR} $: A 3D Neural Additive Model for Interpretable Shape Representation." The Twelfth International Conference on Learning Representations.

---

### Official Review · Reviewer_UJ45 · 2024-11-02

**Soundness:** 3
**Presentation:** 2
**Contribution:** 3
**Rating:** 6
**Confidence:** 3

**Summary:**

Unsupervised disentanglement of transformations and content is a challenging task that was previously approached primarily through using separate ad-hoc transformation methods, or by self-supervised contrastive-based methods. Ad-hoc transformations suffer from being limited to the given parameterization chosen, while self-supervised methods do not tackle this disentanglement problem directly. In this work, DualContrast is proposed, which consists of a VAE with additional contrastive losses designed to disentangle content and transformation. The hardest challenge is obtaining positive pairs of samples with respect to transformations: changing the content while keeping the transformation constant. In this work, this has been done by decoding two random samples from the prior of the transformation latent space while feeding different permutations of the content latent representation to obtain similar transformations with different content. The method is applied to MNIST, LineMod, Starmen Shapes, and Cryo-ET subtomograms with positive results.

**Strengths:**

Originality.
In this work, a novel method to address the problem of creating positive pairs of transformations under content change has been proposed. The core of this work is original.

Quality.
The method proposed was evaluated on a sufficient number of datasets. Although not very complex, they could suffice in showing the potential for this approach. The baselines chosen are also relevant to the method proposed.

Clarity.
The figures in the experiment section allow for a quick qualitative assessment of the performance of the methods. The method explanation is quite clear.

Significance.
This work and the proposed method have shown some potential for successful applications.

**Weaknesses:**

The explanation of the method in the abstract and introduction is especially unclear. This is also a problem because Figure 2 fails to properly and intuitively show the method. Reading the method section explains this more. To improve, I would suggest clearly highlighting  the role of the latent space in the creation of the positive pair that would otherwise be impossible. This could be done similarly to how it was done in Figure 3. Figure 2 would then become useful. Additionally, Figure 2 lacks proper annotations such as labeling of all elements present, and proper caption explaining what happens in the figure in a more complete way. There is some inconsistency in how things are called. In the figure, style, and content are mentioned, however, in the text it is clear that "style" is supposed to be "transformation", please pick one and stick with it in the whole manuscript, either one would suffice, however, transformation is likely to be more accurate.

The contributions are a bit bold. The first contribution, especially, is more context for the work than a contribution and could be removed entirely. Please consider reworking the contributions to be more reflective of the actual content.

The related work section should expand a bit more on the protein part, which is currently very unclear for somebody who is not a practitioner. Please provide more examples, even referring to the appendix to understand the data and the context better.

The method section has a few mistakes and the explanation is very wordy, which makes it hard to follow. I wrote a few observations in the questions section of this review.

The manuscript should include the limitations of this method, especially regarding the latent space-based approach to creating positive transformation pairs. For example, the limitations should address whether this approach could be extended to real-world datasets or whether this approach should be limited to specific types of datasets.

The experiments lack in quantitative results. Although disentanglement is very hard to measure, given the ability to choose datasets, it would be much more convincing to have datasets where a quantitative assessment is possible either in the form of direct supervision (similar to what the disentanglement metric is currently doing), or through some downstream tasks where the disentanglement would be useful. Such tasks could be segmentation, or visual question answer.
The choice of the "human deformation" as a dataset is very confusing, and the results reported are also very underwhelming. Although the generated shapes are better, the data appears to be very trivial, so some more information on the training and the difficulty of fitting such samples would be more convincing. The number of parameters, ablation performed, and results from more baselines would be a step in the right direction. It is especially important to keep including all baselines for all datasets used, the results on human deformation appear to be unfinished. If so, it would have been better to simply exclude the dataset from the manuscript.
Additionally, plots of the latent space are only available for the cellular dataset

**Questions:**

In the method section, condition 1 is very confusing. I think it was meant to be "for all $T \in T$ and $x \in X, h_c(T(x)) = h_c(x)$", but please correct me if I misunderstood this.

In the method section, many terms are used seemingly interchangeably, such as "latent space", "factor", "representation", "transformation", "content". Please clarify these terms. For example, at line 206, "transformation" is used, however, I think it was meant to be "transformation representation" or "transformation factor", unless I misunderstood.

There are a few grammatical and syntactical mistakes, such as inconsistencies in the use of uppercase and lowercase, sometimes writing "shape focused" while other times "shape-focused".

---

> ### Comment · Reviewer_UJ45 · 2024-11-28
>
> The authors have addressed most of my concerns in the general comment and the revised version of the manuscript. I have increased my score.

---

> ### Author Response · Authors · 2024-11-28
> **Individual Response to Reviewer UJ45 (Part 1)**
>
> Dear Reviewer UJ45
>
> We very much appreciate your effort in thoroughly reviewing our paper and providing valuable suggestions. We are glad that you find our work “original” with a “clear” explanation of methods, our choice of baselines “relevant,” and overall our method to have “potential.”
>
> **We greatly appreciate you reviewing the global response and increasing your score.**  We also prepared a point-by-point response to your concerns in case you find some remaining concerns. The response is as follows:
>
> >”The explanation of the method in the abstract and introduction is especially unclear. This is also a problem because Figure 2 fails to properly and intuitively show the method. Reading the method section explains this more. To improve, I would suggest clearly highlighting the role of the latent space in the creation of the positive pair that would otherwise be impossible. This could be done similarly to how it was done in Figure 3. Figure 2 would then become useful. Additionally, Figure 2 lacks proper annotations such as labeling of all elements present, and proper caption explaining what happens in the figure in a more complete way. There is some inconsistency in how things are called. In the figure, style, and content are mentioned, however, in the text it is clear that "style" is supposed to be "transformation", please pick one and stick with it in the whole manuscript, either one would suffice, however, transformation is likely to be more accurate.”
>
> Thanks for pointing out this issue and providing suggestions. We have modified Fig. 2 accordingly (please see our revised manuscript). Now, Fig. 2 clearly visualizes the contrastive pair creation strategy. Consequently, we felt the previous Fig. 3 showing contrastive pair creation with a batch of MNIST digits to be optional and moved it to the Appendix as Fig. 9. We also added more details in the caption explaining the figure. We have resolved the style-transformation inconsistency issue and used the term transformation consistently in the Figure.
>
> >” The contributions are a bit bold. The first contribution, especially, is more context for the work than a contribution and could be removed entirely. Please consider reworking the contributions to be more reflective of the actual content.”
>
> Thanks for your feedback. Based on your suggestion, we reworked the contributions (please see the revised manuscript) and removed the first contribution.
>
> >”The related work section should expand a bit more on the protein part, which is currently very unclear for somebody who is not a practitioner. Please provide more examples, even referring to the appendix to understand the data and the context better.”
>
> Due to the page limit, we could not elaborate on the related work in the main manuscript. We provided additional discussions on Appendix Section A1. According to your suggestion, we referred to this section from the related work section of our main manuscript.
>
> > “The method section has a few mistakes and the explanation is very wordy, which makes it hard to follow. I wrote a few observations in the questions section of this review.”
>
> We have addressed your questions.
>
> > “The manuscript should include the limitations of this method, especially regarding the latent space-based approach to creating positive transformation pairs. For example, the limitations should address whether this approach could be extended to real-world datasets or whether this approach should be limited to specific types of datasets.”
>
> Thanks for your suggestion. We have added a Discussions & Limitations section (Section 5) to the main manuscript and discussed your concerns in that section.

---

> ### Author Response · Authors · 2024-11-28
> **Individual Response to Reviewer UJ45 (Part 2)**
>
> >”The experiments lack in quantitative results. Although disentanglement is very hard to measure, given the ability to choose datasets, it would be much more convincing to have datasets where a quantitative assessment is possible either in the form of direct supervision (similar to what the disentanglement metric is currently doing), or through some downstream tasks where the disentanglement would be useful. Such tasks could be segmentation, or visual question answer. The choice of the "human deformation" as a dataset is very confusing, and the results reported are also very underwhelming. Although the generated shapes are better, the data appears to be very trivial, so some more information on the training and the difficulty of fitting such samples would be more convincing. The number of parameters, ablation performed, and results from more baselines would be a step in the right direction. It is especially important to keep including all baselines for all datasets used, the results on human deformation appear to be unfinished. If so, it would have been better to simply exclude the dataset from the manuscript. Additionally, plots of the latent space are only available for the cellular dataset.”
>
> We added additional quantitative results in Table 1. We included a new metric called SAP score and defined it in the evaluation part. Given the insignificance of human deformation in the context, we excluded them from our revised manuscript. We calculated the metrics across all baselines of all the remaining datasets (Please see Table 1). Given the page limit, latent space plots for only the protein dataset were provided in the main manuscript (Fig. 5), where it has the most significance. We included latent space plots for other datasets in Appendix Sec A4.
>
> The answers to your questions are below:
>
> >Q: “In the method section, condition 1 is very confusing. I think it was meant to be "for all and ", but please correct me if I misunderstood this.”
>
> Yes, you are correct. To remove confusion, we updated the wording in the two conditions in Section 3.1. Instead of “for any $x \in \mathcal{X}$,” we used “$\forall x \in \mathcal{X}$” to be consistent with our wording.
>
> >Q: “In the method section, many terms are used seemingly interchangeably, such as "latent space", "factor", "representation", "transformation", "content". Please clarify these terms. For example, at line 206, "transformation" is used, however, I think it was meant to be "transformation representation" or "transformation factor", unless I misunderstood.
>
> Thanks for pointing this out. We have resolved this confusion in the revised manuscript. In the revised manuscript, we only used factor when referring to the ground truth generative factor of the data. So transformation factor means the actual transformation generative factor of the data. On the other hand, when referring to the encoder-predicted content or transformation, we use the word latent codes or code. We use the term latent space when we refer to all the latent codes for the entire data space. When we only say “content” or “transformation,” we mainly refer to the factor.
>
> >Q: “There are a few grammatical and syntactical mistakes, such as inconsistencies in the use of uppercase and lowercase, sometimes writing "shape focused" while other times "shape-focused".
>
> Thanks for bringing this issue to our attention. We have corrected the inconsistencies as much as possible. We also changed all “shape-focused” in the previous manuscript to “shape-focused” in the revised manuscript. In the revised manuscript, we consistently used “shape focused.”

---

### Official Review · Reviewer_B3mr · 2024-11-07

**Soundness:** 2
**Presentation:** 2
**Contribution:** 2
**Rating:** 5
**Confidence:** 4

**Summary:**

This paper proposes an unsupervised disentangling method to disentangle content and transformation of the input. Specifically, this paper first proposes two conditions that disentanglement of content and transformation should satisfy. Then, this paper proposes a method to construct positive and negative samples with respect to both content and transformation. The key idea is to utilize a variational autoencoder to construct these samples. The experiments are conducted on four datasets, i.e., three of them (mnist, linemod, and starmen) are pure images, and one is protein subtomogram. One quantitative result and several qualitative results are shown to prove the effectiveness of the proposed method.

**Strengths:**

1. This paper proposes well-defined conditions for the disentanglement of content and transformation.
2. The experiments are conducted on four datasets, and comprehensive qualitative results are shown.

**Weaknesses:**

The main concern of this paper is evaluation, which is insufficient and less significant.
1. The first three datasets (mnist, linemod, starmen) are somehow toy datasets, which is less significant in real-world applications.
2. I agree that protein conformation is one meaningful real-world application, but other than map visualizations, it fails to produce convincing evaluation results.
3. There lacks some widely used evaluating metrics in Table 1 to demonstrate the application of the disentanglement.
4. This paper also does not provide comparisons with other baseline methods or state-of-the-art methods.

--------------------------------------

Post-rebuttal:
Thanks for the authors' response. The revision includes more quantitative results and qualitative comparisons, which is appreciated. These additions partially resolved my concerns. However, I noticed that these compared methods are still up to date in 2022, which cannot be considered state-of-the-art to some extent.

Another issue to mention is that the metrics D(c|c) and D(c|z) were used in the initial submission, and the protein experiments are the most important one (in my personal view); it is usually discouraged not to report the main quantitative results of the critical experiments in the initial submission, but added during rebuttal.

Considering the rebuttal and revision, I raised my rating to 5. Please use it sparsely.

**Questions:**

Please refer to the Weaknesses section.

---

> ### Author Response · Authors · 2024-11-28
> **Individual Response to Reviewer B3mr (Updates in Evaluation)**
>
> Dear reviewer B3mr,
>
> We thank you for finding our qualitative results comprehensive and our proposed conditions for content-transformation disentanglement to be well-defined.
>
> Please see our response to your concerns below:
>
> >”The first three datasets (mnist, linemod, starmen) are somehow toy datasets, which is less significant in real-world applications.”
>
> We first evaluated our method and the baselines for these datasets to assess whether they could disentangle conformations from compositions in our protein dataset. Although toy-like, these datasets feature transformations akin to protein conformations, with subtle pixel-level changes. Moreover, the baseline methods- Harmony, SpatialVAE, and VITAE- were all tested on the MNIST dataset, so we also started our experiments with MNIST. LineMod was a reasonable RGB single-object dataset to evaluate since the size of the dataset and the type of transformations are comparable to protein datasets. Starmen may be a bit redundant; consequently, we excluded this dataset in our revised manuscript (as mentioned in the global response).
>
>
> >”I agree that protein conformation is one meaningful real-world application, but other than map visualizations, it fails to produce convincing evaluation results.”
>
> We have provided additional results (Fig. 6 in the revised manuscript) showing how DualContrast can identify distinct conformations of proteins with subtle conformational changes from a protein mixture cryo-ET subtomogram dataset. This could not be achieved with any other method. Given the importance of identifying distinct conformations of proteins in diagnosis and drug discovery, this is undoubtedly a significant contribution. Please go through Fig. 6 and its description (in blue text) from line 490 to line 502 in the revised manuscript.
> We also included quantitative disentanglement results for the protein dataset in Table 1. However, the qualitative results (Fig. 5 and Fig. 6) are more important for this use case.
>
> > "There lacks some widely used evaluating metrics in Table 1 to demonstrate the application of the disentanglement."
>
> We added an additional evaluation metric (SAP score)  in Table 1, which demonstrated the separateness of the latent codes, apart from their informativeness (which is measured with $D_{score}$). We agree that there are many evaluating metrics for disentanglement, but as we mentioned in our manuscript (line 342), all these evaluation metrics have been found to be highly correlated (Locatello et al. [1]). The relevant baseline works, e.g., VITAE [2], Harmony [3], etc., used only these metrics to measure content and transformations’ disentanglement. Consequently, we use only $D_{score}$ and SAP score as evaluation metrics. Since the other metrics are highly correlated, the performance is expected to be similar with different metrics.
>
> > “This paper also does not provide comparisons with other baseline methods or state-of-the-art methods.”
>
> We respectfully disagree with this statement. Throughout our result section, in Table 1, in Fig. 3, Fig. 4, Fig. 5, Fig. 6, in Appendix Fig. 10, Fig. 11, Fig. 12 (numbers based on the revised manuscript)), we provided extensive comparisons with the baseline methods. Our baseline methods include the state-of-the-art unsupervised content-transformation disentangling methods, e.g., Harmony [3], SpatialVAE [4], VITAE [2] , etc. We did not include general disentangled representation learning methods like $\beta$-TC-VAE, Factor-VAE, etc, as they are not specifically designed for content and transformation disentanglement. Moreover, previous works [2,3,4] found that these approaches perform poorly in disentangling content and transformations with their generic strategy.
>
> If you have any method in mind that you want us to compare with, please let us know specifically. We would be happy to compare those methods through experimentation or logical discussion.
>
>
> **References**
>
> [1] Locatello, Francesco, et al. "Challenging common assumptions in the unsupervised learning of disentangled representations." international conference on machine learning. PMLR, 2019.
>
> [2] Skafte, Nicki, and Søren Hauberg. "Explicit disentanglement of appearance and perspective in generative models." Advances in Neural Information Processing Systems 32 (2019).
>
> [3] Uddin, Mostofa Rafid, et al. "Harmony: a generic unsupervised approach for disentangling semantic content from parameterized transformations." Proceedings of the IEEE/CVF Conference on Computer Vision and Pattern Recognition. 2022.
>
> [4] Bepler, Tristan, et al. "Explicitly disentangling image content from translation and rotation with spatial-VAE." Advances in Neural Information Processing Systems 32 (2019).

---

> > ### Author Response · Authors · 2024-12-02
> > **Response Reminder to Reviewer B3mr**
> >
> > Dear reviewer B3mr,
> >
> > We have addressed your initial concerns with the revised manuscript and our global and individual responses above. Since the deadline for reviewer comments to authors is today, please let us know if you have further concerns we can address. If you have no more concerns, we would greatly appreciate your reconsidering your initial score.

---

### Author Response · Authors · 2024-11-28
**Global Response**

We thank the area chairs and reviewers for their efforts in reviewing our paper and providing suggestions. Their helpful comments have greatly enhanced our work.

We have revised the manuscript according to the reviewers' suggestions and marked them in blue in the new version of the manuscript. The main differences include:

1. **We included an additional Fig. 6 demonstrating convincing application-specific evaluation results of DualContrast.** The figure clearly shows how DualContrast can identify subtle conformational changes from a protein mixture cryo-ET subtomogram dataset, highlighting its significance in real-world applications.

2. **We included additional evaluation metrics in Table 1.** In the table, we also reported values for all the evaluation metrics for the protein subtomogram dataset.

3. **We added a Discussions & Limitations section (Section 5)**, discussing when DualContrast is expected to work and when not. We observed that the method disentangles the transformations, causing small pixel-space changes, e.g., subtle conformational changes in proteins, viewpoint changes in LineMod, etc. Identifying subtle changes is vital in scientific image datasets, and our method applies to such cases. However, it is not expected to disentangle transformations causing significant pixel-space changes or transformations not present in the dataset, which is also not feasible in a completely unsupervised manner. We discussed these issues in the newly added section.

4. As suggested by Reviewer UJ45, **we excluded the StarMen dataset** and all its results from the manuscript to maintain consistency of evaluation across all the datasets and the page limit. Moreover, all the reviewers regarded the dataset as insignificant, so we used the space to demonstrate highly significant subtomogram results (Fig. 6).

5. **We changed Fig. 2 to make it more comprehensive and self-explanatory**. Consequently, we felt that the previous Fig. 3, which showed contrastive pair-making for a batch of MNIST digits, was optional in the main manuscript, so we moved it to the Appendix as Fig. 9.

---

### Author Response · Authors · 2024-12-04
**Clarifying Remarks**

Dear reviewers and chairs,

We again greatly appreciate your time and effort in reviewing our work and providing suggestions. Since the deadline for authors' comments is today, we would like to mention several clarifying remarks regarding the work. This may resolve several confusions regarding the work.

**Type of contribution- Fundamental vs Incremental**: We view our work as a fundamental contribution rather than an incremental one. In cryo-EM/ET, many incremental works focus on tasks like particle picking or determining the location of protein complexes from raw images, aiming to improve state-of-the-art performance. In contrast, our work addresses a novel task: disentangling content and transformations where the transformations, such as conformational changes, lack a well-defined parametric form. This is similar to works like SpatialVAE [1], which introduced the task of disentangling content from 2D rotations and translations, and Harmony [2], which extended this to disentangling content from parameterized transformations, including 2D and 3D affine transformations. By addressing this new challenge, our work expands the boundaries of what is achievable in the field.

**Takeaway from cryo-ET results**: The main takeaway from the cryo-ET part of this paper is the demonstration that protein complexes with varying compositions and conformations can be identified from a collection of cryo-ET subtomograms in an unsupervised manner- a capability that previous methods lacked. Earlier approaches were limited to either identifying a few distinct protein complexes from a collection of images or analyzing a few conformations of a single protein complex within a dedicated dataset. However, when presented with collections containing multiple distinct protein complexes and multiple conformations, these methods could, at best, identify only a subset of the complexes and failed entirely to distinguish their conformations. This limitation is clearly illustrated in Fig. 6.


**Apart from cryo-ET, use of simple datasets- MNIST and LineMod**: We set out to disentangle the composition (semantic content) and conformation (transformations causing subtle voxel-level changes) of protein complexes in cryo-ET subtomogram datasets by framing the problem as an unsupervised content-transformation disentanglement task. These datasets present protein complexes in diverse 3D poses, with varying compositions and conformations, making the task inherently complex. To address this, we adopted a strategy of starting with simpler datasets, such as MNIST and LineMod (please note starmen was excluded in the revised paper), which share similar problem contexts. This strategy is consistent with prior works. For instance, SpatialVAE [1] and Harmony [2], while targeting content-transformation disentanglement in cryo-EM and cryo-ET, first validated their methods on MNIST. Similarly, NAISR [3], designed for interpretable shape analysis in medical imaging, initially tested its approach on the starmen dataset to establish its effectiveness on simpler cases. Following this strategy, we demonstrated our method’s success on MNIST and LineMod before applying it to cryo-ET subtomograms for the final experiments.

[1] Bepler, Tristan, et al. "Explicitly disentangling image content from translation and rotation with spatial-VAE." Advances in Neural Information Processing Systems 32 (2019).

[2] Uddin, Mostofa Rafid, et al. "Harmony: a generic unsupervised approach for disentangling semantic content from parameterized transformations." Proceedings of the IEEE/CVF Conference on Computer Vision and Pattern Recognition. 2022.

[3] Jiao, Yining, et al. "$\texttt {NAISR} $: A 3D Neural Additive Model for Interpretable Shape Representation." The Twelfth International Conference on Learning Representations.

- ICLR 2025 Conference Submission7350 Authors

---

### Meta-Review · Area_Chair_QVss · 2024-12-22

**Metareview:**

This paper utilizes a VAE with dual latent spaces for disentangling of content and transformations of cellular 3D protein images. Apart from standard VAE loss, the novel part is contrastive learning loss based on positive and negative pairs in terms of content and transformation. The method here is unsupervised, and those positive and negative pairs are generated without supervision or labels. Clustering of content and transformation codes clearly shows the effectiveness of the method as shown in Fig. 6 in the revised version.
However, reviewers were concerned about real-world impact on downstream tasks. Clustering of latents is another way of visualization for the latents, but not really an application with a profound impact. (There might be a knowledge gap for the reviewers and AC who do not work on biological science).
Another weakness is the lack of comparison to more recent methods on disentangled representation learning as mentioned by the reviewer. Another metric was added in the rebuttal, but overall the amount of quantitative results seems insufficient to be convincing.
The way to construct positive pairs and negative pairs is highly unreliable. Randomly selecting pairs of samples can lead to positive pairs with the same content. Also, it is not clear how to bootstrap the joint network to produce synthetic images with the same content but different transformations, in particular at the beginning of training when the network cannot produce images with high fidelity.

The authors are encouraged to improve the empirical significance of the method and resubmit it to the next conference or to a more domain-specific venue where the impact can be better appreciated.

**Additional Comments On Reviewer Discussion:**

Reviwer B3mr and aQPG were concerned about lack of real-world application of disentanglement and lack of evaluation.
During the rebuttal period, the authors reported one more metric and included clustering as an application in the revised version.

---

### Decision · Program_Chairs · 2025-01-22

Reject